# Classroom-Inspired Multi-Mentor Distillation with Adaptive Learning Strategies

## Abstract

We propose **ClassroomKD**, a novel multi-mentor knowledge distillation framework inspired by classroom environments to enhance knowledge transfer between student and multiple mentors. Unlike traditional methods that rely on fixed mentor-student relationships, our framework dynamically selects and adapts the teaching strategies of diverse mentors based on their effectiveness for each data sample. ClassroomKD comprises two main modules: the **Knowledge Filtering (KF)** Module and the **Mentoring** Module. The KF Module dynamically ranks mentors based on their performance for each input, activating only high-quality mentors to minimize error accumulation and prevent information loss. The Mentoring Module adjusts the distillation strategy by tuning each mentor's influence according to the performance gap between the student and mentors, effectively modulating the learning pace. Extensive experiments on image classification (CIFAR-100 and ImageNet) and 2D human pose estimation (COCO Keypoints and MPII Human Pose) demonstrate that ClassroomKD outperforms existing knowledge distillation methods for different network architectures. Our results highlight that a dynamic and adaptive approach to mentor selection and guidance leads to more effective knowledge transfer, paving the way for enhanced model performance through distillation.

## 1 Introduction

Knowledge distillation (KD) (Hinton et al., 2015) is a widely adopted model compression technique in deep learning, where a smaller, more efficient student model learns to replicate the behavior of a larger, more complex teacher model. While traditional KD methods typically employ a single teacher, multi-teacher (or multi-mentor) distillation has been proposed to further enhance performance by leveraging an ensemble of teachers (You et al., 2017). This setup is expected to provide richer and more diverse knowledge, improving the student's generalization and robustness. We use the term **mentor** to describe all networks involved in teaching the student, regardless of their size or role.

Despite its potential benefits, multi-mentor distillation faces several significant challenges:

**Large Capacity Gap**: Employing multiple large mentors can create a substantial capacity gap between the collective representation power of the mentors and that of the student. This gap can hinder the student's ability to effectively mimic the combined knowledge of the mentors, leading to suboptimal learning outcomes. To bridge this gap, some works (Mirzadeh et al., 2019; Son et al., 2021) have introduced intermediate-sized mentors alongside a large teacher. However, smaller mentors may be less effective, potentially introducing additional errors into the student's knowledge.

**Error Accumulation**: The lower performance of smaller mentors can contribute to cumulative errors in the distillation process. This is particularly problematic in sequential distillation frameworks like TAKD (Figure 1(b)), where each mentor teaches only the subsequent smaller model. Such setups can lead to an "error avalanche," where inaccuracies from lower-performing mentors degrade the student's performance (Son et al., 2021). Although DGKD (Figure 1(c)) attempts to mitigate this by allowing each mentor to teach all smaller models and randomly dropping some mentors during training, these strategies can result in valuable information loss and reduced learning efficiency.

**Lack of Dynamic Adaptation**: The performance gap between the student and its mentors is not static; it evolves throughout training. Current methods do not adequately address these dynamic

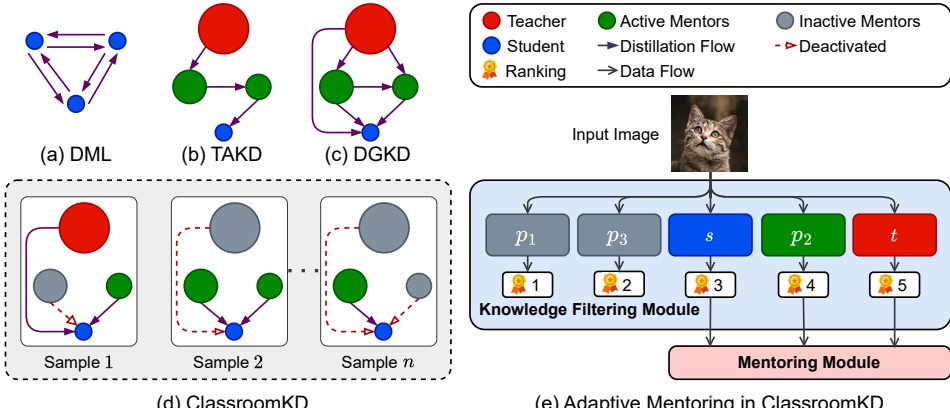

**Figure 1:** (a) DML: Peer models learn from each other without a hierarchical teacher structure. (b) TAKD: A sequential mentor-student hierarchy with large-to-small knowledge transfer. (c) DGKD: Each mentor teaches all smaller models. (d) ClassroomKD: Our proposed method dynamically selects mentors for each data sample based on the current input and ranks them using the Knowledge Filtering Module. (e) Adaptive Mentoring: The Mentoring Module adjusts teaching strategies according to dynamic rankings, ensuring optimal knowledge transfer.

scenarios, limiting the effectiveness of multi-mentor distillation (Hao et al., 2024). Without an adaptive strategy, the potential benefits of multi-mentor distillation are not fully realized.

Observing that **(1)** a mentor's performance varies across different data samples, **(2)** each mentor possesses distinct teaching capabilities due to varying capacity gaps, and **(3)** the performance gap evolves during training, we propose **ClassroomKD** (Figure 1(d)), a novel multi-mentor distillation framework inspired by classroom dynamics (see Appendix E). Our method introduces two key modules (Figure 1(e)) designed to address the following questions:

> **Q1: Which mentors are effective teachers for a given data sample?**
>
> We introduce the **Knowledge Filtering Module** to intelligently select mentors. This module dynamically ranks all mentors based on their performance for each input, activating only those with sufficient performance. A mentor is deemed effective and activated if its predictions are accurate and more confident than the student's. This minimizes error accumulation and information loss.

> **Q2: How much information should the student learn from each active mentor?**
>
> Our **Mentoring Module** addresses this by tuning the teaching strategy based on the performance gap between the student and each active mentor. Specifically, we adjust each mentor's distillation temperature to control the teaching pace, allowing the student to appropriately weigh information received from each mentor before integrating it into its own knowledge.

By addressing these questions iteratively, ClassroomKD ensures a continuously optimized learning process that adapts to the student's evolving capabilities. Our **contributions** are as follows:

1. **ClassroomKD Framework**: We introduce ClassroomKD, a novel multi-mentor distillation framework to dynamically select effective mentors and adapt teaching strategies.
2. **Knowledge Filtering Module**: We develop a Knowledge Filtering Module to enhance distillation quality by selectively activating high-performance mentors, thereby reducing error accumulation and preventing information loss.
3. **Mentoring Module**: We create a Mentoring Module that dynamically adjusts teaching strategies based on the performance gap between the student and each active mentor, optimizing the knowledge transfer process.
4. **Empirical Validation**: Through extensive experiments on image classification (CIFAR-100 and ImageNet) and 2D human pose estimation (COCO Keypoints and MPII Human Pose), we demonstrate that ClassroomKD significantly outperforms state-of-the-art KD methods.

## 2 RELATED WORK

### 2.1 KNOWLEDGE DISTILLATION APPROACHES

Knowledge distillation (KD) (Hinton et al., 2015) is a widely adopted technique for compressing deep neural networks, where a smaller student model learns from a larger teacher model by minimizing the distance between their output probability distributions, or soft labels. Traditional KD methods primarily focus on **logit-based distillation**, where the student learns directly from the teacher's output logits. Notable methods include PKT (Passalis & Tefas, 2018), which employs probabilistic knowledge transfer, FT (Kim et al., 2018), which transfers factorized feature representations, and AB (Heo et al., 2019), which leverages activation boundaries formed by hidden neurons.

**Feature-based distillation** methods transfer knowledge by aligning intermediate representations between the teacher and student. FitNets (Adriana et al., 2015) introduced this approach using intermediate feature maps for training. Later methods like AT (Zagoruyko & Komodakis, 2016), VID (Ahn et al., 2019), and CRD (Tian et al., 2020) enhance knowledge transfer by matching attention maps, utilizing variational information distillation, and employing contrastive learning, respectively.

**Relation-based methods** focus on preserving the structural relationships within the teacher's feature maps. RKD (Park et al., 2019) maintains data point structures through relational knowledge distillation, while SP (Tung & Mori, 2019) and SRRL (Yang et al., 2021) optimize for similarity-preserving objectives. DIST (Huang et al., 2022) addresses large capacity gaps by applying a correlation-based loss to maintain both inter-class and intra-class relationships, enhancing distillation efficiency.

Recent approaches have explored more specialized distillation techniques. WSLD (Zhou et al., 2021) introduces weighted soft labels to balance bias-variance trade-offs, while One-to-All Spatial Matching KD (Lin et al., 2022) focuses on spatial matching techniques. OFA (Hao et al., 2024) optimizes feature-based KD by projecting features onto the logit space, significantly improving performance for heterogeneous models. To enhance distillation effectiveness, several methods have incorporated adaptive strategies. CTKD (Li et al., 2023) dynamically adjusts the temperature during training to gradually increase learning difficulty, and DTKD (Wei & Bai, 2024) employs real-time temperature scaling to improve knowledge transfer efficiency.

### 2.2 MULTI-TEACHER KNOWLEDGE DISTILLATION

Multi-teacher distillation methods aim to further enhance student performance by leveraging an ensemble of mentors (You et al., 2017).

**Online knowledge distillation** has been particularly successful in this context. Deep Mutual Learning (DML) (Zhang et al., 2018) introduces a framework where multiple peer models learn from each other simultaneously during training, fostering collaborative learning among smaller networks and outperforming traditional one-way (offline) distillation. Other online methods include ONE (Zhu et al., 2018), OKDDip (Chen et al., 2020), and FFM (Li et al., 2020), which often outperform offline methods. Online distillation has also been extended to pose estimation tasks (Li et al., 2021b). SHAKE (Li & Jin, 2022) proposed using proxy teachers with shadow heads to use the benefits of online distillation in offline settings.

To address the **capacity gap** in multi-teacher setups, Teacher-Assistant KD (TAKD) (Mirzadeh et al., 2019) employs intermediate-sized teacher assistants (TAs) to bridge the gap between the largest teacher and the student. However, sequential distillation through TAs can result in an "error avalanche," where errors propagate at each step, reducing final performance. Adaptive Ensemble Knowledge Distillation (AEKD) (Du et al., 2020) mitigates this issue by using an adaptive dynamic weighting strategy to reduce error propagation in the gradient space. Densely Guided KD (DGKD) (Son et al., 2021) further improves upon these methods by guiding each TA with both larger TAs and the main teacher, enabling a more gradual and effective transfer of knowledge. Additionally, DGKD introduces a strategy of randomly dropping mentors during training to expose the student to diverse learning sources, enhancing overall learning robustness.

While existing multi-teacher methods offer various mechanisms for knowledge distillation, they still grapple with challenges such as managing the capacity gap, mitigating error accumulation, and adapting to dynamic mentor-student relationships.

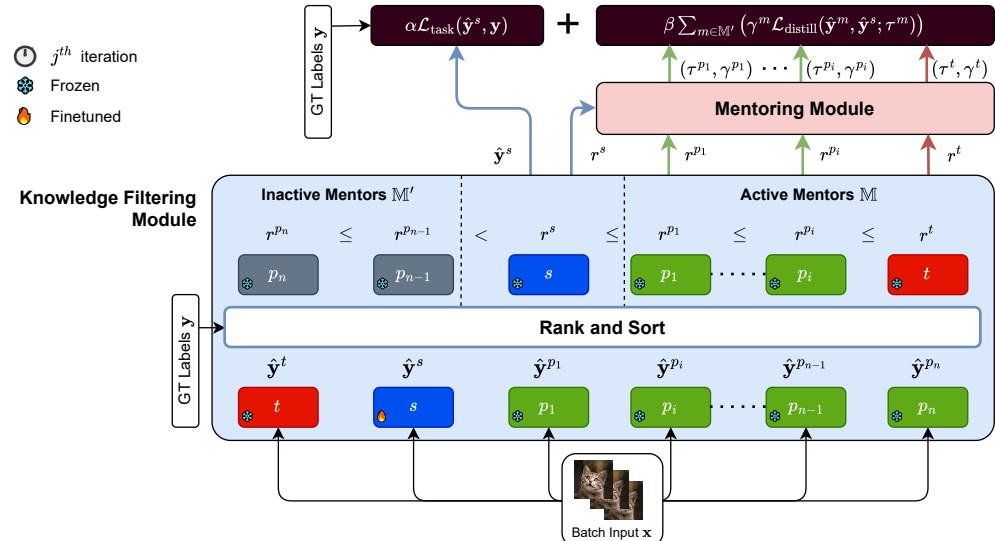

Figure 2: **The ClassroomKD framework.** comprises a **Knowledge Filtering (KF) Module** and a **Mentoring Module**. The KF Module optimizes learning by selectively incorporating feedback from higher-ranked mentors, reducing noise transfer and preventing error accumulation. The Mentoring Module adjusts mentor influence based on their performance relative to the student.

## 3 METHODOLOGY

ClassroomKD is a novel multi-mentor distillation framework inspired by real-world classroom environments. It is designed to address the challenges of large capacity gaps, error accumulation, and lack of dynamic adaptation. Our framework is illustrated in Figure 2.

**Classroom Definition.** A classroom comprises **(1)** a high-capacity *teacher* model, $t$, **(2)** a small *student* model, $s$, and **(3)** $n$ *peer* models of intermediate capacities, $\mathbb{P} = \{p_i\}_{i=1}^n$. We define $\mathbb{M} = \{t\} \cup \mathbb{P}$ as the set of pre-trained mentors that remain frozen during the student's training process. At each training step, the student distills knowledge from a dynamically selected subset of mentors, called the *active mentors* ($\mathbb{M}' \subseteq \mathbb{M}$). The set of all classroom models is denoted $\mathbb{C} = \{s\} \cup \mathbb{M}$. We use the Knowledge Filtering (KF) Module for intelligent mentor selection and the Mentoring Module to adjust the teaching pace based on the capacity gap of each mentor-student pair.

### 3.1 KNOWLEDGE FILTERING MODULE

The KF Module is designed to intelligently select which mentors should contribute to the student's learning process for each data sample. This selective approach mitigates error accumulation and prevents the student from learning from less effective mentors.

Let $\boldsymbol{x} = \{x_k\}_{k=1}^N$ be a batch of training data with size $N$, and $\boldsymbol{y} = \{y_k\}_{k=1}^N$ be the ground-truth labels. The batch inputs $\boldsymbol{x}$ are forwarded through all classroom models to obtain the predicted logits $\hat{\boldsymbol{y}}^m$, which are then converted to probabilities with a softmax operation. We isolate the probability assigned to the true class $\boldsymbol{y}$ and compute a weighted average of the correct prediction probability across the batch for each model. For all $m \in \mathbb{C}$, this is defined as:

$$\hat{\boldsymbol{y}}^m = m(\boldsymbol{x}) \tag{1}$$

$$\boldsymbol{p}^m = \text{softmax}(\hat{\boldsymbol{y}}^m) \tag{2}$$

$$\boldsymbol{p}_{\text{gt}}^m = \boldsymbol{p}^m[\boldsymbol{y}] \tag{3}$$

$$w^m = \frac{1}{N} \sum_{k=1}^N \boldsymbol{p}_{\text{gt}}^m(x_k) \tag{4}$$

The weights $w^m$ reflect the performance of model $m$ on the current training batch. We use the computed weights as a proxy for mentor suitability in the distillation process and rank mentors based

on their relative performance to all classroom models:

$$r^m = \lambda \left( \frac{w^m}{\sum_{m \in \mathbb{C}} w^m} \right) \tag{5}$$

where $r^m$ is a normalized ranking score of model $m$, and $\lambda$ is a scaling parameter set to the number of mentors in the classroom. Active mentors $\mathbb{M}'$ are defined as those with higher ranks than the student:

$$\mathbb{M}' = \{m \mid m \in \mathbb{M} \text{ and } r^m > r^s\} \tag{6}$$

This ensures the student learns from high-quality sources by selecting mentors based on their performance ranks. This selective approach **prevents error accumulation** as only mentors outperforming the student can teach it, avoiding the propagation of errors from less effective mentors. Additionally, it **avoids information loss** by consistently selecting the best-performing mentors, unlike random mentor-dropping strategies (Son et al., 2021).

### 3.2 MENTORING MODULE

The Mentoring Module dynamically adjusts the influence of each active mentor based on the mentor-student performance gap. This **adaptive teaching strategy** facilitates effective knowledge transfer tailored to the student's evolving ability to absorb information from each mentor.

The distillation loss minimizes KL divergence between the student and mentor's output distributions:

$$\mathcal{L}_{distill}(P, Q; \tau) = \tau^2 \cdot \mathrm{KL} \left( \mathrm{softmax}\left(P/\tau\right) \,\|\, \mathrm{softmax}\left(Q/\tau\right) \right) \tag{7}$$

where $P$ and $Q$ represent the logits from the mentor and student networks, respectively, and $\tau$ is a temperature hyperparameter that smooths the probability distributions during the distillation process.

The temperature $\tau$ controls the sharpness of the probability distributions, affecting the knowledge transfer from a mentor to the student. For each active mentor $m \in \mathbb{M}'$, we adjust the distillation temperature $\tau^m$ based on the performance gap between the student and the mentor. The performance gap is measured as the difference in their ranking scores:

$$\Delta r^m = |r^m - r^s|/r^m \tag{8}$$
$$\tau^m = 1 + \Delta r^m \cdot \tau \tag{9}$$

Here, $\tau$ is the base temperature, and $\tau^m$ increases with $\Delta r^m$, which represents the mentor-student performance gap. A larger $\Delta r^m$ results in a higher $\tau^m$, smoothing the mentor's output distribution. This adjustment theoretically slows down the distillation process by softening the mentor's predictions, allowing the student to assimilate knowledge more gradually when the performance gap is large. Conversely, the student receives sharper, more direct guidance when the gap is small.

The total loss $\mathcal{L}$ is computed by combining a task-specific loss $\mathcal{L}_{\text{task}}$ with the weighted distillation losses from all active mentors:

$$\mathcal{L} = \alpha \mathcal{L}_{\text{task}}(\hat{\boldsymbol{y}}^s, \boldsymbol{y}) + \beta \sum_{m \in \mathbb{M}'} \gamma^m \mathcal{L}_{\text{distill}}(\hat{\boldsymbol{y}}^m, \hat{\boldsymbol{y}}^s; \tau^m) \tag{10}$$

Here, $\alpha = r^s$ represents the student's self-confidence, which scales the task-specific loss. As the student's rank $r^s$ improves, $\alpha$ increases, encouraging the student to rely more on its own predictions. For each mentor $m$, $\gamma^m = r^m$ scales the corresponding distillation loss, where $r^m$ is the mentor's rank relative to the student. $\beta$ is a hyperparameter to control the influence of distillation loss relative to the task loss. This weighing, along with the mentor-specific temperature $\tau^m$, ensures that higher-performing mentors have a greater influence on the student's learning, with each mentor distilling knowledge at an appropriate rate based on the performance gap. We use Cross-Entropy Loss for classification and MSE Loss for pose estimation tasks.

This promotes independent learning by increasing the student's reliance on its own task performance as its confidence grows. It also ensures that the student benefits from guidance based on the relative performance of the active mentors, effectively balancing task-specific training with distillation from the most suitable mentors. This dynamic and adaptive approach ensures **optimized knowledge transfer**, minimizes error accumulation, and enhances the overall performance of the student model.

## 4 EXPERIMENTS

This section presents our experiments to evaluate the effectiveness of ClassroomKD using different datasets. We primarily use CIFAR-100 (Krizhevsky et al., 2009) classification for detailed comparisons with state-of-the-art single and multiple-teacher distillation methods. This also includes online approaches using multiple mentors. In addition, we also report results on ImageNet (Deng et al., 2009) classification and human pose estimation using the COCO Keypoints (Lin et al., 2014) and MPII Human Pose (Andriluka et al., 2014) datasets. Our results show that ClassroomKD outperforms existing methods under various settings, highlighting the robustness and adaptability of our method.

**Implementation Details.** For CIFAR-100, we train for 240 epochs with a batch size of 64, a learning rate of 0.05 decayed by 10% every 30 epochs, and a 120-epoch warm-up phase. We use SGD with 0.9 momentum and $5 \times 10^{-4}$ weight decay. The temperature $\tau$ is set to 12 via grid search (Figure 3). For ImageNet, models are trained for 100 epochs with $\tau = 8$. For pose datasets, models are trained for 210 epochs with $\tau = 4$. The scaling factor $\lambda$ is $n + 1$ for all experiments, where $n$ is the number of peers. We used $\beta = 1.0$ for classification and $\beta = 2.5$ for pose estimation. We follow standard training protocols, with mentors pre-trained and kept frozen.

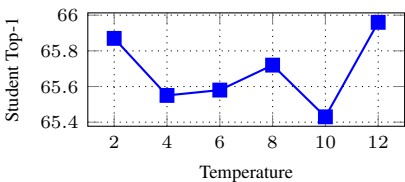

Figure 3: **Temperature selection.** Grid search using fixed-temperature KD, with the best student performance at $\tau = 12$, used as the base temperature in Eq. 9.

### 4.1 RESULTS

**CIFAR-100 Classification:** Table 1 compares the performance of ClassroomKD to *single-teacher distillation methods* on the CIFAR-100 dataset. We evaluate various teacher-student pairings using both homogeneous and heterogeneous architectures. ClassroomKD, which is logit-based, performs better than logit-based methods in a majority of cases, as well as most feature-based and relation-based methods. In particular, when comparing with the recent state-of-the-art CTKD (Li et al., 2023) and DTKD (Wei & Bai, 2024) methods, both of which use adaptive temperatures, ClassroomKD is better than the former in all cases and the latter in 75% of the cases.

Table 1: **Comparison with single-teacher distillation methods on CIFAR-100 classification.** We report top-1 accuracy (%). KD methods are grouped by feature, relation, and logit-based. Best values in logit-based methods are **bold**, second-best underlined, and overall best blue

| Method | Homogeneous architectures | | | | Heterogeneous architectures | | | | |
|---|---|---|---|---|---|---|---|---|---|
| Teacher | R110 | R110 | R56 | VGG13 | VGG13 | R32×4 | W-40x2 | R50 | Swin-T |
| Student | R20 | R32 | R20 | VGG8 | MBV2 | SN-V2 | SN-V1 | MBV2 | R18 |
| NOKD | 69.06 | 71.14 | 69.06 | 70.68 | 64.60 | 71.82 | 70.50 | 64.60 | 74.01 |
| FitNets (Adriana et al., 2015) | 68.99 | 71.06 | 69.21 | 73.54 | 64.14 | 73.54 | 73.73 | 63.16 | 78.87 |
| AT (Zagoruyko & Komodakis, 2016) | 70.22 | 72.31 | 70.55 | 73.62 | 59.40 | 72.73 | 73.32 | - | - |
| VID (Ahn et al., 2019) | 70.16 | 72.61 | 70.38 | 73.96 | - | 73.40 | 73.61 | 67.57 | - |
| CRD (Tian et al., 2020) | 71.46 | 73.48 | 71.16 | 73.94 | 69.73 | 75.65 | 76.05 | 69.11 | 77.63 |
| SimKD (Chen et al., 2022) | - | - | - | 74.93 | - | 77.49 | - | - | - |
| SMKD (Lin et al., 2022) | 71.70 | 74.05 | 71.59 | 74.39 | - | - | - | - | - |
| RKD (Park et al., 2019) | 69.25 | 71.82 | 69.61 | 73.72 | 64.52 | 73.21 | 72.21 | 64.43 | 74.11 |
| SP (Tung & Mori, 2019) | 70.04 | 72.69 | 69.67 | 73.44 | 66.30 | 74.56 | 74.52 | - | - |
| SRRL (Yang et al., 2021) | 71.51 | 73.80 | - | 73.23 | 69.34 | 75.66 | 76.61 | - | - |
| DIST (Huang et al., 2022) | - | - | 71.75 | - | - | 77.35 | - | 68.66 | 77.75 |
| KD (Hinton et al., 2015) | 70.67 | 73.08 | 70.66 | 72.98 | 67.37 | 74.45 | 74.83 | 67.35 | 78.74 |
| PKT (Passalis & Tefas, 2018) | 70.25 | 72.61 | 70.34 | 73.37 | - | 74.69 | 73.89 | 66.52 | - |
| FT (Kim et al., 2018) | 70.22 | 72.37 | 69.84 | 73.42 | - | 72.50 | 72.03 | - | - |
| AB (Heo et al., 2019) | 69.53 | 70.98 | 69.47 | 74.27 | - | 74.31 | 73.34 | - | - |
| WSLD (Zhou et al., 2021) | 72.19 | 74.12 | 72.15 | - | - | 75.93 | 76.21 | - | - |
| CTKD (Li et al., 2023) | 70.99 | 73.52 | 71.19 | 73.52 | 68.46 | 75.31 | 75.78 | 68.47 | - |
| DTKD (Wei & Bai, 2024) | - | 74.07 | 72.05 | 74.12 | 69.01 | 76.19 | **76.29** | 69.10 | - |
| OFA (Hao et al., 2024) | - | - | - | - | - | - | - | - | 80.54 |
| Ours (2024) | **72.45** | **74.60** | **72.65** | **74.51** | **69.84** | **76.52** | 75.05 | **70.15** | 80.32 |

**CIFAR-100 Classification with Multiple Mentors.** We compare our approach with online and offline strategies involving multiple mentors in Table 2a. In addition to the teacher named in the table,

**Table 2: Comparison with multi-teacher distillation methods.** Best and second-best values in offline methods are **bold** and underlined, respectively, and overall best in blue. *:trained on (coco+aic)

**(a) Results on CIFAR-100 classification.** We report top-1 accuracy (%). KD methods are grouped by online and offline. ClassroomKD is offline.

| Method | Same Architectures | | | | Mixed Architectures | |
|---|---|---|---|---|---|---|
| Teacher | WR40x2 | R110 | R56 | VGG13 | VGG13 | W-40x2 |
| Student | WR16x2 | R20 | R20 | VGG8 | MBV2 | SN-V1 |
| NOKD | 73.64 | 69.06 | 69.06 | 70.68 | 64.60 | 70.50 |
| DML (Zhang et al., 2018) | 74.83 | 70.55 | 70.24 | 72.86 | 66.30 | 74.52 |
| ONE (Zhu et al., 2018) | 74.68 | 70.77 | 70.43 | 72.01 | 66.26 | - |
| SHAKE (Li & Jin, 2022) | 75.78 | - | 71.62 | 73.85 | 68.81 | 76.42 |
| TAKD (Mirzadeh et al., 2019) | 75.04 | - | 70.77 | 73.67 | - | - |
| AEKD (Du et al., 2020) | 75.68 | 71.36 | 71.25 | **74.75** | 68.39 | 76.34 |
| EBKD (Kwon et al., 2020) | | | | 74.10 | 68.24 | 76.61 |
| DGKD (Son et al., 2021) | 76.24 | - | 71.92 | 74.40 | - | - |
| CA-MKD (Zhang et al., 2022) | - | - | - | 74.30 | 69.41 | **77.94** |
| AVER | 74.98 | 71.20 | 71.08 | 73.18 | 62.94 | 73.00 |
| Ours | **76.51** | **72.45** | **72.65** | 74.51 | **69.84** | 75.18 |

**(b) Results on ImageNet.**

| T: RG-Y320, 4 Peers | | | |
|---|---|---|---|
| Student | NOKD | AVER | Ours |
| R32 | 73.31 | 74.60 | **75.20** |

**(c) Results on Pose Estimation** with four mentors. We report PCKh for MPII and AP for COCO.

| Dataset | MPII | | COCO |
|---|---|---|---|
| Teacher | HRNet-W32-D | | RTMP-L* |
| Student | LiteHRNet-18 | | RTMPose-t |
| Peers | Same | Mixed | Same |
| NOKD | 85.91 | 85.91 | 68.20 |
| AVER | 86.64 | 86.07 | 69.26 |
| Ours | **86.72** | **86.37** | **69.73** |

our classroom uses five additional peers, as defined in Appendix A. ClassroomKD performs better than online methods like DML, ONE, and SHAKE, as well as the baseline offline method (AVER) in almost all cases. Our method also outperforms methods like TAKD, AEKD, and DGKD, which are specifically designed to address the capacity gap and error accumulation issues. In particular, the AEKD method achieves significantly poor results despite using four more mentors than us. We also have a comparable performance with the state-of-the-art CA-MKD method for networks they report.

**ImageNet Classification with Multiple Mentors:** In the ImageNet experiments, ClassroomKD demonstrates its scalability and effectiveness on a larger dataset. As shown in Table 2b, ClassroomKD outperforms traditional KD and other multi-teacher methods, achieving higher top-1 and top-5 accuracy scores. This indicates that ClassroomKD maintains its superiority even as the complexity and size of the dataset increase, underscoring its robustness and adaptability.

**Pose Estimation with Multiple Mentors.** We also assess ClassroomKD's performance on 2D human pose estimation tasks using the COCO Keypoints and MPII Human Pose datasets. Table 2c compares ClassroomKD against a simple multi-teacher baseline regarding keypoint detection accuracy and overall pose estimation performance. ClassroomKD achieves higher Average Precision (AP) scores, demonstrating its ability to effectively transfer structured knowledge from multiple mentors to the student model. This highlights ClassroomKD's versatility and effectiveness beyond image classification tasks, extending its applicability to complex, structured prediction problems.

## 4.2 ABLATION STUDIES

We conduct a series of ablation studies to understand the individual contributions of different components of our ClassroomKD framework, providing insights into our design choices.

**Table 3: Ablation study** to assess the contribution of different components of ClassroomKD.

**(a) Role of Multiple Mentors.** Single-teacher distillation slightly improves student performance compared to vanilla training. Adding intermediate mentors (peers) and using adaptive distillation further enhances learning.

| Student | Teacher | Peers | Adaptive Distillation | Top-1 Accuracy |
|---|---|---|---|---|
| ✓ | ✗ | ✗ | ✗ | 63.31 |
| ✓ | ✓ | ✗ | ✗ | 63.35 |
| ✓ | ✓ | ✓ | ✗ | 65.96 |
| ✓ | ✓ | ✓ | ✓ | **68.52** |

**(b) Adaptive Distillation in ClassroomKD.** We analyze the role of the KF Module and Mentoring Module in our adaptive method. Both components contribute to overall performance.

| KF Module | Mentoring Module | Top-1 Accuracy |
|---|---|---|
| ✗ | ✗ | 65.96 |
| ✗ | ✓ | 67.25 |
| ✓ | ✗ | 68.49 |
| ✓ | ✓ | **68.52** |

**Role of System Components.** In Table 3a, we observe a significant improvement when moving from single-teacher distillation (row 2) to a multi-mentor setup (row 3). The presence of multiple mentors,

specifically the intermediate-sized peers, bridges the capacity gap between the large teacher and the small student. This gap is a well-known limitation in traditional KD, where the student struggles to fully comprehend the knowledge transferred from a much larger teacher. Introducing peers, which have capacities between the teacher and student, effectively provides a smoother learning gradient for the student, facilitating a more gradual and interpretable knowledge transfer.

The **adaptive distillation strategy** (row 4) boosts accuracy by 2.56%, highlighting the limitations of static distillation methods. By adjusting distillation based on the student's progress and mentor outputs, ClassroomKD ensures more efficient learning, especially during critical phases where mentor usefulness varies. Table 3b shows that the KF Module improves accuracy from 65.96% to 68.49% by filtering out irrelevant knowledge, while the Mentoring Module dynamically adapts teaching strategies, raising performance to 67.25%. Together, these modules achieve the highest accuracy of 68.52%, ensuring both quality and adaptability in knowledge transfer.

We examine the classroom composition and further analyze our framework in the following sections.

### 4.2.1 CLASSROOM SIZE AND COMPOSITION

This section examines the impact of both the number and diversity of mentors on student performance within ClassroomKD. Our experiments investigate different mentor configurations, including varying mentor quantities and diverse architectures and performance levels.

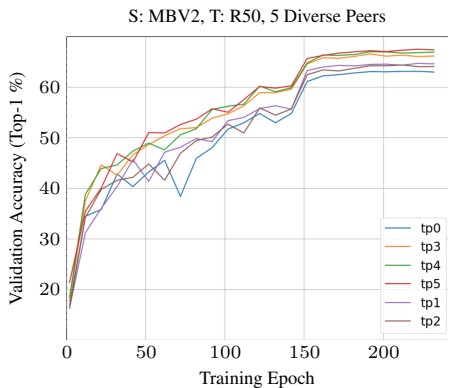

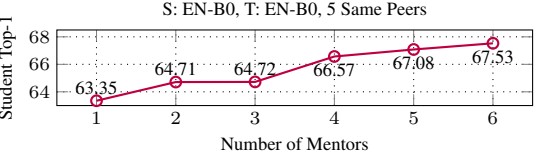

**(b)** Fixing mentor architecture and size by using multiple instances of the same mentor at different training checkpoints, we observe that student accuracy still improves with the number of mentors. This indicates that diversity in mentor performance alone is enough to enhance student learning.

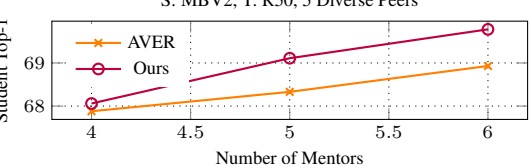

**(a)** Without controlling mentor architectures or performance levels, we train a student in classrooms with up to six mentors. The validation accuracy improves as the number of peers increases, but the marginal gain diminishes beyond five peers. We use these results to limit the size of our classrooms to six mentors in all subsequent experiments.

**(c)** Comparing our approach to vanilla multi-mentor distillation (AVER) highlights the benefit of our adaptive distillation with dynamic mentor selection as the classroom grows.

**Figure 4: Effect of Classroom Size and Composition.** We investigate how the number of mentors, their architectures, and performance differences affect learning.

**Impact of peer quantity.** Figure 4a and 4b illustrates the effect of increasing the number of peers in the classroom. Without any peers, the student achieves 63.35% top-1 accuracy. However, as peers are added, performance steadily improves, reaching 67.53% with five peers. This improvement demonstrates that incorporating intermediate mentors (peers) with varied capacities helps bridge the gap between the large teacher and small student, making knowledge transfer more effective. However, the performance improvement plateaus beyond five peers. This suggests that while adding mentors benefits learning, the gain diminishes beyond a certain point due to redundancy in the knowledge being transferred. Therefore, we limit our classrooms to six mentors in all subsequent experiments to balance efficiency and performance.

**Architectural Diversity** (Table 4a): We observe that using mentors with diverse architectures (e.g., VGG, ResNet, and ShuffleNet) yields better performance (68.52%) compared to using multiple instances of the same architecture (67.53%). Interestingly, this improvement occurs despite the fact that the total parameter count of the diverse mentors (12.3M) is significantly lower than that of the

**Table 4: Effect of Mentor Diversity.** We investigate the role of mentor diversity in terms of architecture and performance levels.

**(a) Diversity in mentor architectures.** Using diverse mentor architectures improves distillation performance compared to a homogeneous setup, even when the total parameter count of the diverse mentors is lower. This indicates that architectural diversity provides valuable learning signals.

**(b) Diversity in mentor performance.** Classrooms with low-performing mentors, average mentors (a mix of medium and high performers), and diverse mentors (a combination of low, medium, and high performers) are compared. The diverse group, with a balanced mix of performance levels, yields the best student accuracy, highlighting the benefit of including mentors with varied accuracy for effective distillation.

| Classroom | Mentors | Params | Top-1 |
|---|---|---|---|
| Same | EN-B0 x6 | 24.8M | 67.53 |
| Diverse | VGG13, R8, R14, R20, SV1, SV2 | **12.3M** | **68.52** |

| Mentors | 20-50% | 50-65% | 65-73% | Top-1 |
|---|---|---|---|---|
| Low | ✓✓✓ | ✓✓ | - | 67.77 |
| Average | - | ✓✓✓ | ✓✓ | 67.53 |
| Diverse | ✓ | ✓✓ | ✓✓ | **68.29** |

homogeneous set (24.8M). This indicates that architectural diversity introduces richer and more varied learning signals, which are more effective for knowledge distillation.

**Performance Diversity** (Table 4b): We also evaluate the effect of mentor performance diversity by creating classrooms composed of mentors from different performance brackets. When mentors are homogeneous in terms of performance (either all low- or all high-performing), student performance remains lower. However, a diverse set of mentors, comprising both low- and high-performing peers, leads to the highest student accuracy (68.29%). This suggests that having varied knowledge sources across performance levels provides complementary learning experiences for the student, facilitating more robust distillation.

### 4.2.2 TEMPERATURE IN MENTORING MODULE

We explore the role of adaptive temperature ($\tau$) in the Mentoring Module and its impact on bridging the capacity gap between classroom networks. Our approach adjusts the temperature dynamically based on the student's learning progress, with higher $\tau$ values at the start to accommodate the larger capacity gap, which gradually decreases as the student's understanding improves. This adaptive strategy allows mentors to effectively "slow down" the teaching process during early stages and accelerate it later, ensuring effective knowledge transfer.

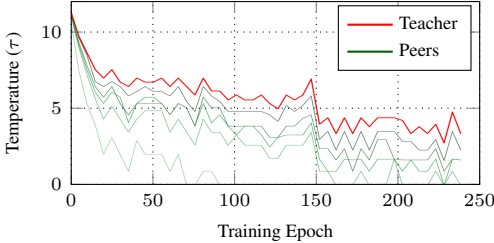

**Figure 5: Effect of temperature adaption.** Our adaptive approach independently adjusts the temperature for each mentor (teacher and peers) over time, allowing them to optimize their teaching strategies dynamically across epochs.

**Table 5: Temperature adaption strategy.** We compare our temperature adaptation method to DTKD (Wei & Bai, 2024) by replacing our mentoring module with their dynamic temperature computation. Our mentoring module outperforms DTKD's temperature adaption strategy with $\tau = 12$ (our default) and $\tau = 4$ (tuned for DTKD).

| Method | Adaption | $\tau$ | MBV2 | R20 |
|---|---|---|---|---|
| DTKD | DTKD | 4 | 69.10 | 72.05 |
| Ours | DTKD | 4 | 64.36 | 71.18 |
| Ours | DTKD | 12 | 68.03 | 70.02 |
| Ours | Ours | 12 | **70.15** | **72.65** |

In our experiments, using an adaptive $\tau$ strategy yields a significant improvement in student performance. The adaptive method, which adjusts $\tau$ based on the student's progress, achieves a top-1 accuracy of 69.78%, compared to a static $\tau$ setup where performance remains lower (65.43% to 65.87% for fixed values). This demonstrates that adapting the teaching pace based on the student's understanding leads to better learning outcomes.

**Comparison with DTKD.** We compared our approach with DTKD's dynamic temperature strategy by adding their method to our mentoring module. While DTKD works well with a single teacher (row 1), it is not as effective when used with multiple mentors of different capabilities. This is because

DTKD assumes that all mentors predict the correct label and does not fully address the dynamic capacity gap between the teacher and student during the training process. In contrast, our method masks mentor logits with ground-truth labels, and adapts more effectively to evolving capacity gaps, achieving consistently better results across different network architectures.

### 4.2.3 RANKING STRATEGIES IN KF MODULE

We study the effect of our ranking strategy in the KF Module, which dynamically activates the teacher and peers to guide the student. In Figure 6, we observe the evolution of ranks over time, where the teacher (red) consistently holds a higher rank than all other mentors because of its superior performance. Peer ranks (green) fluctuate, and ineffective peers are deactivated as their ranks fall below the student's rank (blue) during training. This dynamic mentor activation prevents error accumulation from underperforming mentors and allows the student to progressively improve.

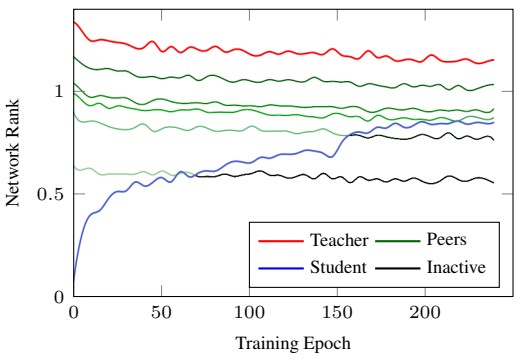

**Figure 6: Rank-based mentor activation.** Ranks evolve during training, reflecting the dynamic nature of capacity gaps. ClassroomKD uses high-quality mentors (red and green), deactivating ineffective mentors (black) who rank below the student (blue).

**Table 6: Choice of Ranking Strategy.** We compare three ranking methods. Here, we employ different networks for peers. (A) we use class probabilities as $\alpha$, $\beta$. (B) we employ class ranks as $\alpha$, $\beta$. (C) We use a dynamically calculated $\lambda$. We observe that using ranks as loss weights improves student network performance compared to probabilities.

| Teacher
Student | WR40x2
WR16x2 | R110
R20 | R110
R32 | R56
R20 | VGG13
VGG8 |
|---|---|---|---|---|---|
| Method B | 75.42 | 71.94 | 74.28 | 72.56 | 73.58 |
| Ours | **76.51** | **72.45** | **74.60** | **72.65** | **74.51** |
| Teacher
Student | VGG13
MBV2 | R32x4
SN-V2 | R32x4
SN-V1 | R50
MBV2 | WR40x2
SN-V1 |
| Method B | 68.52 | 75.71 | **75.08** | 69.78 | **75.96** |
| Ours | **69.84** | **76.52** | 74.84 | **70.15** | 75.05 |

In Table 6, we explore an alternative ranking strategy (**Method B**) by replacing Eq. 5 with:

$$\boldsymbol{j} = \text{argsort}(w^m \mid m \in \mathbb{C}) \quad \text{for } m \in \mathbb{C} \tag{11}$$

$$r^m = \lambda \cdot \boldsymbol{j}^{-1}(m) \tag{12}$$

where $r^m$ is a ranking score, $\lambda$ is a scaling parameter set to 0.1, and $\boldsymbol{j}^{-1}(m)$ gives the index of model $m$ in a sorted list of weights. This results in uniformly distributed ranks (0.1, 0.2, 0.3, ...) instead of the weighted rank distribution in our original formulation. The results show that the proposed ranking method works better. However, we note that even this alternative ranking computation performs better than baseline methods for multiple networks. This improvement stems from the rank-based weighting mechanism, which focuses the student's learning on more challenging and discriminative classes, reducing sensitivity to noise and enhancing overall learning efficiency.

## 5 CONCLUSION

We presented *ClassroomKD*, a novel knowledge distillation framework that mimics a classroom environment, where a student learns from a diverse set of mentors. By selectively integrating feedback through the Knowledge Filtering (KF) Module and dynamically adjusting teaching strategies with the Mentoring Module, ClassroomKD ensures effective knowledge transfer and mitigates the issues of error accumulation and capacity gap. Our approach significantly improves the student's performance in classification and pose estimation tasks, consistently outperforming traditional distillation methods.
**Limitations.** While we demonstrated the efficacy of ClassroomKD on image classification and human pose estimation, its application to other domains and more complex tasks, such as object detection and segmentation, presents a promising avenue for future work. Despite the improvements, the framework introduces complexity, especially with respect to the mentor ranking and teaching adjustments, which can require careful tuning. Future work will explore further optimizations and expand the framework's utility to broader tasks.

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

# A  TRAINING PROTOCOLS

**Mentor Configuration.** We use a predefined order for the mentor set in all experiments for consistency. Any deviations from this are clearly stated.

**Table 7: Mentor Configurations.** We show the set of models used in all our experiments along with their respective top-1 accuracies and the ensemble performance. The size of the mentors, if all the peers were replaced by the teacher $((n+1))t$), the size of the current mentors ($1tnp$), and student size are also mentioned. Model abbreviations: MB: MobileNet, SN: ShuffleNet, R: Resnet, W: WRN, EN: EfficientNet, SQ: SqueezeNet, RP: RTMPose, HR: HRNet, LHR: LiteHRNet, RG: RegNet

| | | | Mentors | | | | | Params (M) | |
|---|---|---|---|---|---|---|---|---|---|
| $s$ | $t$ | $p_1$ | $p_2$ | $p_3$ | $p_4$ | $p_5$ | $(n+1)t$ | $1tnp$ | $s$ |
| **CIFAR-100 Classification** | | | | | | | | | |
| R20 (69.06) | R110 (74.31) | R8 (60.22) | R14 (67.28) | SN-V2 (72.60) | MBV2 (63.51) | SN-V1 (71.29) | 10.42 | 5.12 | 0.27 |
| R32 (71.14) | R110 (74.31) | R8 (60.22) | R14 (67.28) | SN-V2 (72.60) | MBV2 (63.51) | SN-V1 (71.29) | 10.42 | 5.12 | 0.47 |
| R20 (69.06) | R56 (72.41) | R8 (60.22) | R14 (67.28) | SN-V2 (72.60) | MBV2 (63.51) | SN-V1 (71.29) | 5.17 | 4.24 | 0.27 |
| VGG8 (70.36) | VGG13 (74.64) | R20 (69.06) | MBV2 (63.51) | SN-V2 (72.60) | R56 (72.41) | R110 (74.31) | 56.77 | 14.50 | 3.96 |
| MBV2 (63.51) | VGG13 (74.64) | R8 (60.22) | R14 (67.28) | R20 (69.06) | SN-V1 (71.29) | SN-V2 (72.60) | 56.77 | 12.31 | 0.81 |
| SN-V2 (72.60) | R32x4 (79.42) | R8 (60.22) | R14 (67.28) | R20 (69.06) | MBV2 (63.51) | SN-V1 (71.29) | 44.62 | 9.739 | 1.35 |
| SN-V1 (71.29) | W-40-2 (75.61) | R20 (69.06) | MBV2 (63.51) | SN-V2 (72.60) | R56 (72.41) | VGG13 (74.64) | 13.53 | 15.00 | 0.95 |
| MBV2 (63.51) | R50 (79.34) | R8 (60.22) | R14 (67.28) | R20 (69.06) | SN-V1 (71.29) | SN-V2 (72.60) | 142.23 | 26.55 | 0.81 |
| SN-V1 (71.29) | R32x4 (79.42) | R8 (60.22) | R14 (67.28) | R20 (69.06) | MBV2 (63.51) | SN-V2 (72.60) | 44.62 | 10.14 | 0.95 |
| W-16-2 (73.64) | W-40-2 (75.61) | R20 (69.06) | MBV2 (63.51) | SN-V2 (72.60) | R56 (72.41) | VGG13 (74.64) | 13.53 | 14.98 | 0.70 |
| MBV2 (63.51) | ENB0 (73.21) | ENB0 (60.23) | ENB0 (61.03) | ENB0 (63.60) | ENB0 (66.87) | ENB0 (72.70) | 24.81 | 24.81 | 0.81 |
| R18 (74.01) | Swin-T(224) (88.78) | SN-V2 (72.60) | W-40-2 (75.61) | VGG13 (74.64) | R32x4 (79.42) | - | 137.98 | 48.10 | 11.22 |
| **ImageNet Classification** | | | | | | | | | |
| R34 (73.31) | RG-Y320 (80.74) | SQ1-1 (58.18) | MBV2 (71.88) | ENB3 (78.54) | RG-Y016 (77.67) | - | 725.23 | 173.22 | 21.79 |
| **COCO Keypoints Estimation** | | | | | | | | | |
| RP-t (68.2) | RP-1* (76.5) | RP-s (71.6) | RP-m (74.6) | RP-l (75.8) | - | - | - | - | - |
| **MPII Human Pose Estimation** | | | | | | | | | |
| LHR-18 (85.91) | HR-W32D (90.4) | LHR-30 (86.9) | HR-W32 (90.0) | HR-W48 (90.1) | - | - | - | - | - |
| LHR-18 (85.91) | HR-W32D (90.4) | SN-V2 (82.8) | MBV2 (85.4) | R50 (88.2) | - | - | - | - | - |

**Hardware and Software Configuration.** We trained most of our CIFAR-100 experiments on a single V100-16GB GPU. The time required for an experiment ranged between 4 and 4.5 hours on average. We build our code on top of `Image Classification SOTA` repository[1] and MMPose, and use pretrained models from these libraries as our mentors.

---

[1] https://github.com/hunto/image_classification_sota/

# B   FUTURE DIRECTION: CLASSROOMKD AND DATASET DISTILLATION

ClassroomKD shows strong potential in knowledge distillation, and one promising extension is its application in dataset distillation, which can further broaden its impact across various tasks.

Dataset distillation aims to create small, synthetic datasets that enable neural networks to achieve comparable performance to those trained on the original, much larger datasets. This approach reduces computational costs and storage requirements while maintaining model generalization. By optimizing a small set of representative training samples, a distilled dataset $S$ is generated such that a model trained on $S$ performs well on the original dataset $\mathcal{T}$. In our experiments, we use **FRePo** (Zhou et al., 2022) to create a distilled CIFAR-100 dataset, reducing each class to only 10 samples (Figure 7). Of these, 7 images per class are used for training, while the remaining 3 are used for testing.

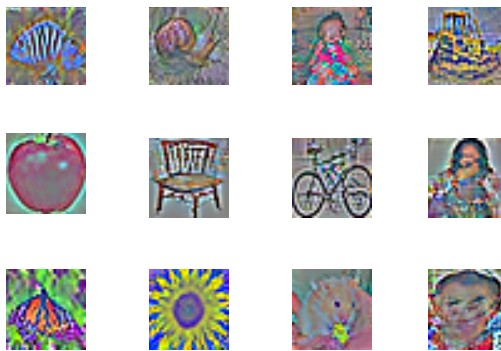

**Figure 7: Sample from the distilled CIFAR-100 dataset created using FRePo.** The dataset is reduced to 10 representative images per class, where each image encapsulates key characteristics of the class. This distilled dataset significantly reduces storage and computational requirements while maintaining essential features for effective training.

As shown in Table 8, we conducted experiments on this distilled CIFAR-100 dataset and evaluated validation performance on the full CIFAR-100 dataset using the MobileNetV2 and ResNet-20 architectures. Notably, the standalone MobileNetV2 student achieves 31.00 on the distilled dataset, with 3.75% top-1 accuracy on the full validation set. However, applying ClassroomKD with 1 teacher and 5 peers significantly improves performance, reaching 44.34 on the distilled data and 6.30% top-1 accuracy on the full CIFAR-100 validation set. This is in stark contrast to the AVER approach, which results in only 2.33 on the distilled data and 1.51% top-1 accuracy on the full validation set using the same number of mentors. Similarly, ClassroomKD achieves superior results with ResNet-20, showing a notable 9.66 percentage point improvement on the distilled data compared to NOKD and a 1.85 percentage point gain on the full CIFAR-100 validation set.

**Table 8: Performance comparison on the distilled CIFAR-100 dataset and validation metrics on the full CIFAR-100 dataset.** Results show top-1 accuracy on both the distilled dataset (7 images per class for training) and the full CIFAR-100 validation set. ClassroomKD (1 teacher, 5 peers) outperforms both the standalone student and AVER, demonstrating its efficacy in low-data regimes.

| Student | MobileNetV2 | | ResNet-20 | | |
|---|---|---|---|---|---|
| Method | Distilled Top-1 | Top-1 | Distilled Top-1 | Top-1 | Top-5 |
| NOKD | 31.00 | 3.75 | 50.00 | 3.08 | 12.50 |
| AVER | 2.33 | 1.51 | 32.00 | 3.55 | 15.24 |
| ClassroomKD | **44.34** | **6.30** | **59.66** | **4.93** | **17.81** |

These results suggest that ClassroomKD has strong potential to enhance performance on compact datasets, even where traditional methods fall short. By selectively leveraging the most effective mentors, ClassroomKD enables optimal knowledge transfer, making it a promising approach for dataset distillation. Additionally, combining ClassroomKD with dataset distillation can be extended to **continual learning**, where models from previous tasks act as mentors for new tasks. This approach could improve efficiency and performance in larger-scale tasks and real-world scenarios.

# C    ANALYSIS AND ADDITIONAL RESULTS

## C.1    PER-CLASS PERFORMANCE IMPROVEMENT

We further analyze ClassroomKD's effectiveness by examining the per-class performance improvements of the distilled student model compared to the baseline model (without knowledge distillation). To this end, we compare the class-level accuracy differences between ClassroomKD and a standard multi-teacher knowledge distillation (AVER) approach, both using the distilled CIFAR-100 dataset.

In Figure 8, we illustrate the performance differences between the ClassroomKD student and the baseline model on the left. ClassroomKD improves performance in 86 out of 100 classes while minimizing performance degradation in the remaining classes. In contrast, AVER (right) has a significantly smaller improvement, and the absolute performance degradation is more severe than with ClassroomKD. This demonstrates the benefit of our mentor ranking strategy, which dynamically selects mentors based on their relative performance and reduces the likelihood of detrimental knowledge transfer or error accumulation from multiple mentors.

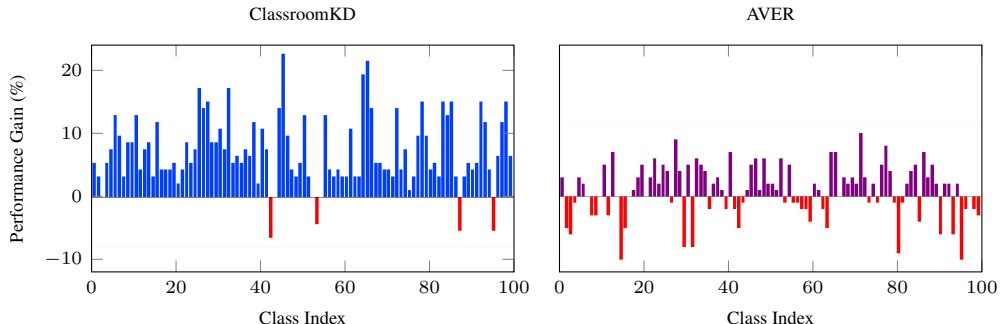

**Figure 8: Comparison of per-class performance gain over the NOKD baseline.** With ClassroomKD (left), the distilled model improves performance on 86 classes. With multi-teacher KD without mentor ranking (right), significantly fewer classes improve, the absolute improvement is smaller, and the remaining classes experience larger performance degradation (red bars). This highlights the impact of our dynamic strategies in improving performance across different classes.

## C.2    SOFTMAX FOR RANKING MENTOR

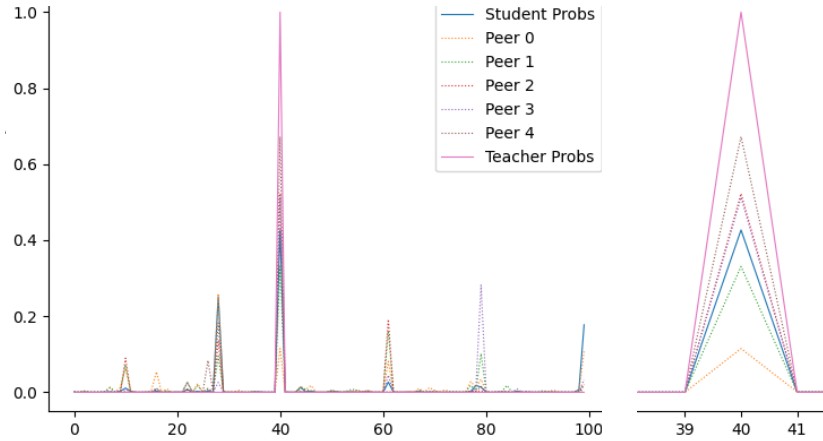

**Figure 10: Softmax for Ranking Mentors.** We plot the softmax probabilities (y-axis) against the class indices (x-axis) of all models in a **1t5p** classroom used. On the left, we show the distribution of confidence in model predictions. We zoom in on the target class on the right. The mentors with less softmax probability (or confidence) than the student (blue line) are deactivated to reduce error accumulation observed in traditional multi-teacher distillation methods.

## C.3 POSE ESTIMATION QUALITATIVE RESULTS

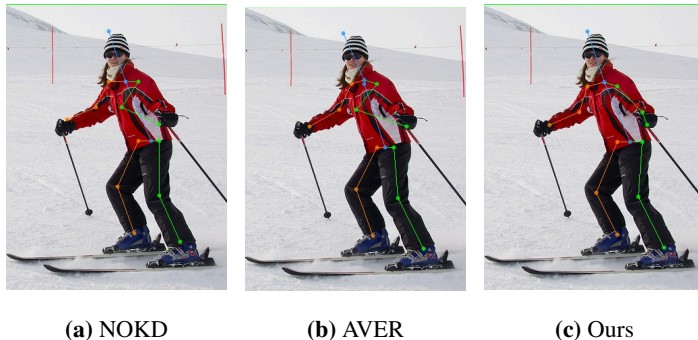

**(a)** NOKD      **(b)** AVER      **(c)** Ours

**Figure 11: Qualitative comparison of LiteHRNet-18 student** trained **(a)** without distillation, **(b)** with six teachers in a naive manner, and **(c)** with six teachers using ClassroomKD.

## D CLASSROOMKD ALGORITHM

---

**Algorithm 1** ClassroomKD

---

**Require:** Input batch $\mathbf{x}$
**Require:** Ground truth labels $\mathbf{y}$
**Require:** Student $s$
**Require:** Mentors $\mathbb{M} \leftarrow \{t\} \cup \{p_i\}_{i=1}^{n}$
**Require:** $\beta$: weight of distillation loss
1: weights $\leftarrow \{\}$        // Initialize empty dictionary for mentor weights
2: ranks $\leftarrow \{\}$        // Initialize empty dictionary for mentor ranks
3: $\mathcal{L} \leftarrow 0$        // Initialize total loss
4: mask $\leftarrow$ onehot($\mathbf{y}$)
5: $\mathbb{C} \leftarrow \{s\} \cup \mathbb{M}$
6: **for** $m \in \mathbb{C}$ **do**
7:     $\hat{\boldsymbol{y}}^m \leftarrow m(\boldsymbol{x})$        // Get predictions from model $m$
8:     $\boldsymbol{p}^m \leftarrow \text{softmax}(\hat{\boldsymbol{y}}^m)$        // Convert logits to probabilities
9:     $\boldsymbol{p}_{\text{gt}}^m \leftarrow \boldsymbol{p}^m \cdot \text{mask}$        // Isolate probabilities assigned to ground truth
10:     $w^m \leftarrow \text{average}(\boldsymbol{p}_{\text{gt}}^m, \text{dim-1})$        // Average correct class probability for model $m$
11:     weights$[m] \leftarrow w^m$        // Store weight for model $m$
12: **end for**
13: weights $\leftarrow$ dict(sorted(weights.items(), key=lambda item: item[1]))   // Sort mentors by weight
14: total_weight $\leftarrow \sum$(weights.values())        // Calculate sum of all mentor weights
15: ranks $\leftarrow \{m : (|\mathbb{M}| \cdot w)/\text{total\_weight for } m, w \in \text{weights.items()}\}$        // Assign rank scores
16: **for** $m \in \mathbb{M}$ **do**
17:     **if** ranks$[m] > $ ranks$[s]$ **then**
18:        $\tau^m \leftarrow \frac{\text{ranks}[m] - \text{ranks}[s]}{\text{ranks}[m]}$
19:        $\mathcal{L}_{\text{distill}} \leftarrow \text{KL}(\hat{\boldsymbol{y}}^m, \hat{\boldsymbol{y}}^s, \tau^m)$
20:        $\mathcal{L}_{\text{distill}} \leftarrow \text{ranks}[m] \cdot \mathcal{L}_{\text{distill}}$
21:     **else**
22:        $\mathcal{L}_{\text{distill}} \leftarrow 0$
23:     **end if**
24:     $\mathcal{L} \leftarrow \mathcal{L} + \mathcal{L}_{\text{distill}}$        // Add distillation loss to total loss
25: **end for**
26: $\mathcal{L}_{\text{task}} \leftarrow \text{CELoss}(\hat{\boldsymbol{y}}^s, \text{targets})$        // Compute task loss (e.g., cross-entropy)
27: $\mathcal{L}_{\text{task}} \leftarrow \text{ranks}[s] \cdot \mathcal{L}_{\text{task}}$        // Weight task loss by student's rank
28: $\mathcal{L} \leftarrow \mathcal{L}_{\text{task}} + \beta \cdot \mathcal{L}$        // Combine task loss and distillation loss
29: **return** $\mathcal{L}$        // Return the total loss

---

# E CLASSROOM LEARNING STYLES SURVEY

We conducted an online survey about learning styles and academic success in the classroom environment, in which forty (40) respondents participated. Most respondents (92.5%) were 18-45 years old, with 32.5% self-identifying as students, 22.5% as teachers or mentors, and 37.5% identifying as both. This survey aimed to gather insights into the various methods and strategies students and teachers employ to excel in their academic goals. In this appendix, we provide some statistics from the responses we received. These inspired the ClassroomKD approach introduced in the paper. Participation in the survey was voluntary, and participants could withdraw at any time without penalty.

**Consent form for the survey**

# Classroom-learning styles

Hello! Welcome to our survey on learning styles and academic success in the classroom environment!

**Research Purpose:**

Learning is a continuous process and happens mostly through our experiences and interactions. A classroom provides a more structured way to understand the world around us. At a young age, we cannot or don't experience a lot of things firsthand (eg. experience snow or see where Lowest Common multiple is applied). It provides a way to learn more abstract concepts and things we can't experience often. The purpose of this survey is to gather insights into the various methods and strategies students and teachers employ to excel in their academic endeavors. By understanding these different perspectives, we aim to translate them into a model in a deep learning setting and study if these strategies work well for neural networks.

**Consent:**

By participating in this survey, you consent to the use of your responses anonymously for research purposes only (for my thesis). Your privacy and confidentiality will be strictly maintained throughout the study. We do not have any age restrictions. However, if you are below the age of 13, a guardian must consent to your participation and assist you in your responses.

**Withdrawal Rights:**

You have the right to withdraw from this study at any point. If you wish to withdraw or have any further queries, please contact us via email a

**Responses:**

Please answer the following questions spontaneously and honestly. There are no right or wrong answers. We are interested in learning what works best for you personally.

Thank you for your participation! Your insights are invaluable for my thesis! Have fun (re)visiting your school days :)

## E.1  ROLE OF A COMPETITIVE CLASSROOM ENVIRONMENT

In the first series of questions, we try to find out if students feel like they learn better in collaborative environments, which provide opportunities for healthy competition. The results showed positive response to collaboration among peers along with the teacher. However, competition was mostly detrimental to learning towards the end of the training period (after the completion of coursework and during their exams).

**How does competition among peers affect your learning abilities?**

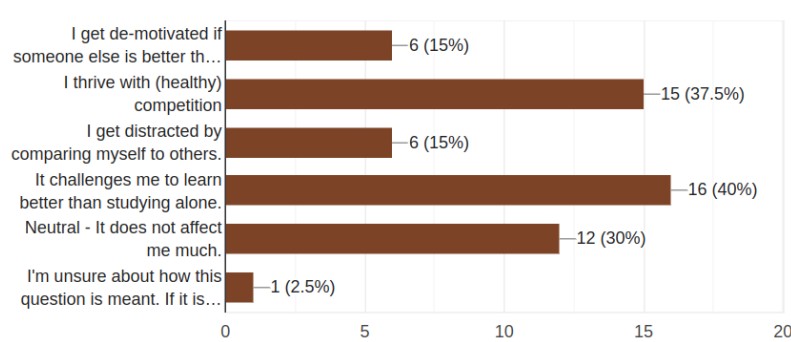

The survey further explored specific scenarios where competition was beneficial or detrimental

**Competition among peers helps me when:**

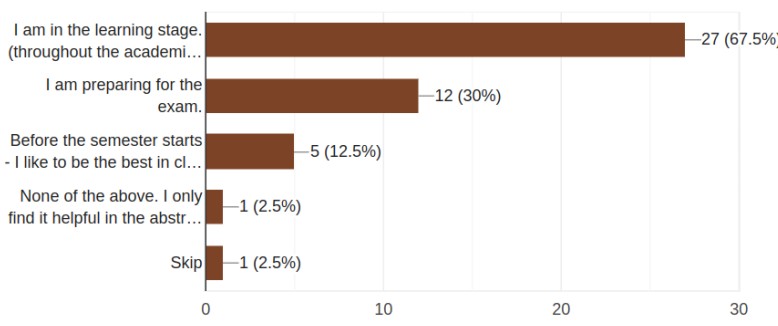

Competition was found to be helpful during the learning phase (lecture period) of a semester. This competition can take the form of in-class discussions, group projects, or other collaborative activities. It encouraged active participation and knowledge sharing among students, fostering a collaborative learning atmosphere.

**Competition among peers is distracting when:**

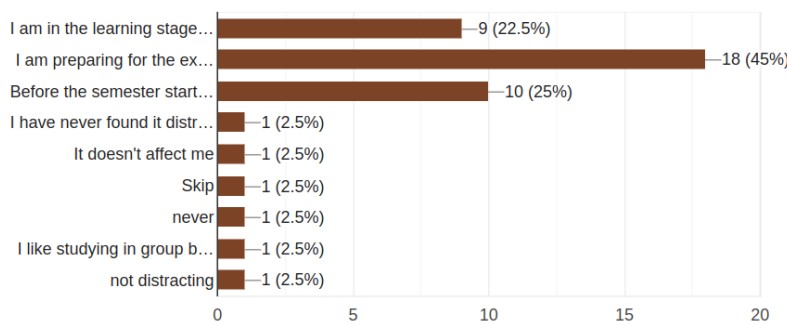

On the other hand, competition was often seen as distracting during critical phases like final exams or major project submissions. In these scenarios, the pressure to outperform peers led to decreased focus and increased anxiety, negatively impacting overall performance.

The insights from these responses were instrumental in designing the ClassroomKD framework. Recognizing the dual nature of competition, we incorporated mechanisms to balance collaborative learning with individual performance enhancement:

- **Collaborative Learning Environment**: By integrating multiple peers in the knowledge distillation process, ClassroomKD emulates a collaborative classroom where the student model benefits from diverse feedback. This mirrors the beneficial aspects of peer competition, fostering a supportive learning environment.
- **Performance-Based Filtering**: To mitigate the negative effects of competition, the Knowledge Filtering Module ensures that the student model learns from higher-ranked mentors only. This selective approach reduces the pressure from underperforming models and prevents the error propagation that could arise from unhealthy competition.

### E.2 SEEKING GUIDANCE

The second set of questions focused on understanding how students seek guidance when faced with challenges and the effectiveness of the feedback received. In these questions, we attempt to understand what prompts students to seek guidance from their mentors and how they handle it. The goal was to understand the correlation between when or whom students are asking for help and their success in achieving their objectives.

**When your confidence drops, whom do you usually ask your doubts?**

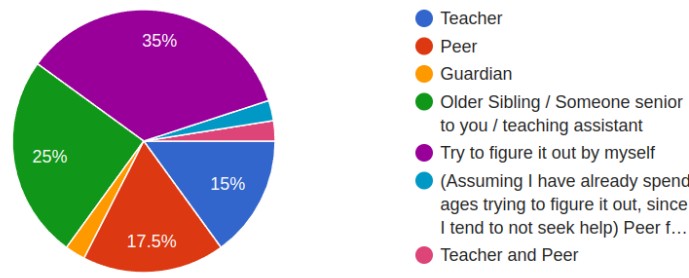

The responses indicated a preference for different sources based on the perceived expertise and approachability. Most respondents consulted their peers or older siblings or tried to figure things out themselves. Peers were considered more approachable and could provide relatable explanations.

**When you asked your questions to your teacher, what was their response?**

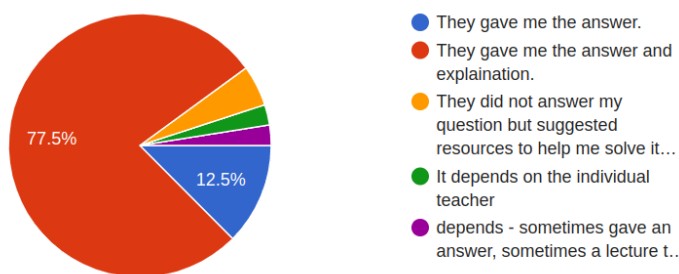

When asked about the nature of the teacher's response, many participants noted that teachers often provided detailed explanations and additional resources. This thorough approach helped clarify doubts and improve understanding.

**Did the teacher's strategy help you gain confidence?**

Many respondents confirmed that their confidence increased after receiving teacher feedback. This highlights the importance of effective mentoring in the learning process.

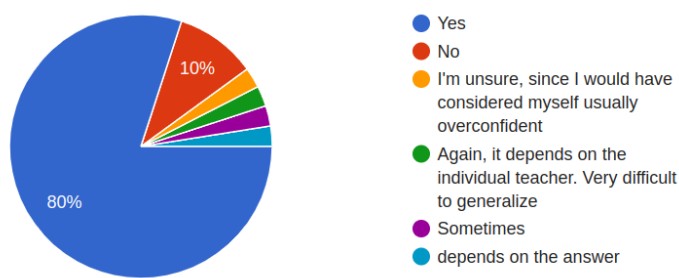

These insights were crucial in shaping the Mentoring Module of ClassroomKD:

- **Adaptive Mentoring**: Inspired by the positive impact of teacher feedback, the Mentoring Module dynamically adjusts the teaching strategies based on the student's current performance level. This ensures that the student model receives guidance tailored to its needs, similar to how a teacher would adjust their approach based on a student's understanding.

- **Selective Feedbac**k: To emulate the preference for high-performing peers, the Knowledge Filtering Module ensures that the student model seeks feedback from higher-ranked peers and teachers. This selective process enhances the quality of knowledge transfer and boosts the student model's confidence over time.

E.3 SELF-ASSESSMENT AND FEEDBACK

The final set of questions aimed to understand how students assess their own performance and the role of feedback in enhancing their learning experience.

**How do you assess your performance on a test?**

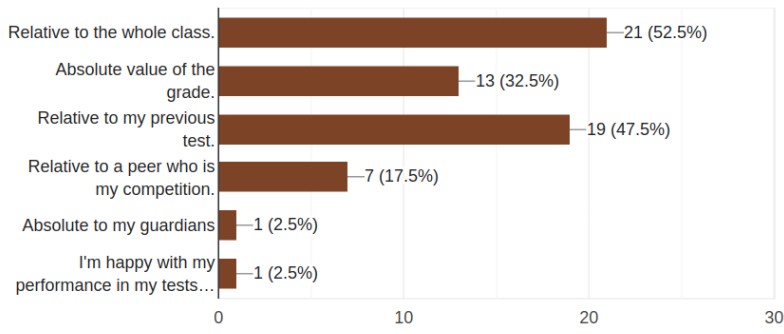

Most of the responses suggest that students assess their performance based on peer comparison.

**My confidence increases when I am appreciated:**

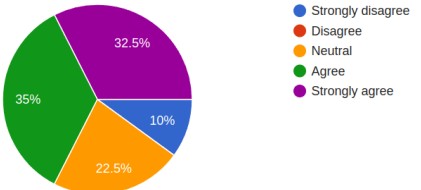

Respondents indicated that appreciation from others significantly boosted their confidence. Positive reinforcement motivated them to continue their efforts and strive for better results.

The responses highlighted the importance of self-assessment and constructive feedback, which influenced the design of ClassroomKD:

- **Progressive Confidence Boosting**: Reflecting the impact of appreciation on confidence, ClassroomKD incorporates a Progressive Confidence Boosting strategy. As the student model's performance improves, its self-confidence (represented by the weighting parameter $\alpha$) increases. This dynamic adjustment ensures that the model's learning is reinforced by its achievements, similar to how students gain confidence from positive feedback.
- **Continuous Improvement**: By integrating detailed feedback mechanisms through the Mentoring Module, ClassroomKD ensures that the student model continuously learns from its mistakes. The adaptive teaching strategies help the student model bridge the performance gap with mentors over time, fostering a continuous improvement cycle.

The survey responses provided valuable insights into effective learning strategies in a classroom environment. These insights were directly translated into the design and implementation of the ClassroomKD framework, ensuring that our knowledge distillation approach mirrors successful educational practices and optimizes student model performance.

# F SUPPLEMENTARY MATERIAL

## F.1 EXPANDED MULTI-MENTOR DISTILLATION COMPARISON

**Designing a Simple Baseline**: We use AVER as the simplest baseline in our multi-mentor comparisons in Tab. 2. This is a direct counterpart of KD in single-teacher experiments and is defined as:

$$\mathcal{L}_{\text{AVER}} = \mathcal{L}_{\text{task}}(\hat{\boldsymbol{y}}^s, \boldsymbol{y}) + \sum_{m \in \mathbb{M}} \mathcal{L}_{\text{distill}}(\hat{\boldsymbol{y}}^m, \hat{\boldsymbol{y}}^s; \tau) \tag{13}$$

Here, each teacher is weighted equally without any ranking or temperature adaption; the student naively attempts to learn the aggregate of all teachers' knowledge. This simple multi-teacher baseline is also used in existing SOTA works, including SHAKE (Li & Jin, 2022) and CA-MKD (Zhang et al., 2022). We present an extended version of our multi-mentor comparison on CIFAR-100 in Tab. 9. This table highlights the improvement of different methods over AVER.

**Table 9: Comparison of our method with various online, sequential and multi-teacher methods.** We report top-1 accuracy (%). Teachers and students are grouped by *same* architecture and *different* architecture. KD baselines are grouped by online and offline. Best results are **bold**, and second-best are underlined. Δ represents accuracy gain of a method over AVER results from the paper where the results were taken from. Rows marked with * correspond to the results taken from SHAKE (Li & Jin, 2022), and †corresponds to the results taken from CA-MKD (Zhang et al., 2022).

| Method | Homogeneous architectures | | | | Heterogeneous architectures | | |
|---|---|---|---|---|---|---|---|
| | WR40x2 (75.61) WR16x2 | R110 (74.31) R20 | R56 (72.34) R20 | VGG13 (74.64) VGG8 | VGG13 (74.64) MN-V2 | W-40x2 (75.61) SN-V1 | |
| NOKD | 73.64 | 69.06 | 69.06 | 70.68 | 64.60 | 70.50 | |
| DML (Zhang et al., 2018)* | 74.83 | 70.55 | 70.24 | 72.86 | 66.30 | 74.52 | |
| ONE (Zhu et al., 2018)* | 74.68 | 70.77 | 70.43 | 72.01 | 66.26 | - | |
| SHAKE (Li & Jin, 2022)* | 75.78 | - | 71.62 | 73.85 | 68.81 | 76.42 | |
| TAKD (Mirzadeh et al., 2019) | 75.04 | - | 70.77 | 73.67 | - | - | |
| DGKD (Son et al., 2021) | 76.24 | - | 71.92 | 74.40 | - | - | |
| **AVER*** | 75.22 | 71.24 | 71.08 | 74.90 | 68.91 | 76.30 | |
| AEKD (Du et al., 2020)* | 75.68 | 71.36 | 71.25 | **74.75** | 68.39 | 76.34 | |
| Δ over AVER* | +0.46 | +0.12 | +0.17 | -0.15 | -0.52 | +0.04 | (avg) +0.02 |
| **AVER†** | - | - | - | 74.07 | 68.91 | 76.30 | |
| EBKD (Kwon et al., 2020)† | - | - | - | 74.10 | 68.24 | 76.61 | |
| Δ over AVER† | - | - | - | +0.03 | -0.67 | +0.31 | (avg) -0.11 |
| AEKD (Du et al., 2020)† | - | - | - | 73.38 | 68.39 | 76.34 | |
| Δ over AVER† | - | - | - | -0.69 | -0.52 | +0.04 | (avg) -0.39 |
| CA-MKD (Zhang et al., 2022)† | - | - | - | 74.30 | 69.41 | **77.94** | |
| Δ over AVER† | - | - | - | +0.23 | +0.5 | +1.64 | (avg) +0.79 |
| **AVER (ours)** | 74.98 | 71.20 | 71.08 | 73.18 | 62.94 | 73.00 | |
| ClassroomKD (ours) | **76.51** | **72.45** | **72.65** | 74.51 | **69.84** | 75.05 | |
| Δ over AVER (ours) | +1.53 | +1.25 | +1.57 | +1.33 | +7.01 | +2.05 | (avg) **+2.45** |

As shown in the table above, our approach surpasses all knowledge distillation baselines in most cases. However, Δ over AVER is significantly higher in our case than other methods in 100% cases.

The AVER results reported in SHAKE and CA-MKD do not align with each other or our AVER. This is because of differences in the types of teachers used. Unlike single-teacher KD, multi-teacher KD is not standardized, and details about which additional teachers were used are scarce. We provide a detailed configuration of each classroom in our experiments in Tab. 7 to overcome this lack of standardization and make it easier for future works to compare their approaches with us.

## F.2   Intuition Behind Proposed Ranking Method

**Classroom Dynamics.** For a given sample $x_k$, we can visualize the output probability distribution of a model $m$ by plotting the softmax probability $P_{z_i}^m$ of its logit $z_i$ against the class labels $i$, for all $i \in C$. The models in a classroom can have logit distributions that fall into one of the three cases: (1) Weak classifiers predict the true label $y_k$ with low confidence. (2) Strong classifiers predict the true class with high confidence, giving it a "sharper" peak. (3) Wrong classifiers have a peak at the wrong class. This is illustrated in Fig. 12.

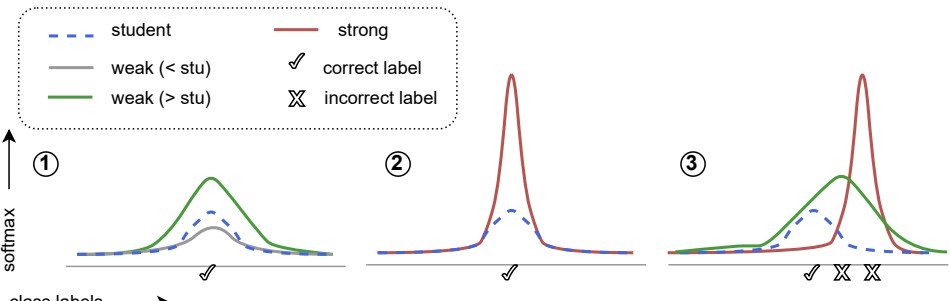

**Figure 12: Illustration of the classroom models' probabilistic distributions** The student encounters three types of mentors while learning: 1. weak classifiers predict with low confidence. 2. strong classifiers are highly confident in their prediction. 3. Wrong classifiers predict incorrect labels.

With this in mind, we can make the following claims about the conditions an ideal knowledge distillation framework must satisfy:

- In case (1), we notice two mentors with correct predictions, one above (green) and another below (gray) the student (blue). As we empirically show in Fig. 6 and Sec. F.2.1, the student probability for the correct class surpasses the weaker mentors as training progresses. Therefore, at any point in time, as long as the student prediction is not incorrect, **it should only learn from those mentors who have a "sharper" peak than itself as the goal of knowledge distillation is to pull the student's peak at the correct class upwards**.

- If we compare cases (1) and (2), despite both mentors predicting the correct class label with higher probability than the student, **the red classifier is "sharper" than the green classifier. Hence, the student must give more importance to the feedback from the red classifier.**

- The ability of the student to approximate the distribution of a teacher is inversely related to the distance between the height of the student and the teacher's peaks. As this distance is larger at the start of training and progressively decreases, **the extent of softening the mentor distribution depends on the gap between the "sharpness" of the mentor and the student and varies with time.**

To address the first point, we can introduce selective distillation by activating the mentors above the student. This selective activation should take into account both the correctness and the relative confidence (probability) of the correct prediction. In our method, this is ensured via Eq. 3 and 5. The second and third points relate to the weight and temperature, respectively, used in each mentor's loss. In our method, these are quantified as the $\gamma^m$ and $\tau^m$ parameters in Eq. 17.

Coincidentally, we can neatly connect all these concepts by calculating the "sharpness" of the models and comparing them against each other. The question arises: **how can sharpness be quantified?** In DTKD (Wei & Bai, 2024), which is similarly motivated as our method, the sharpness is defined as the $\mathrm{logsumexp}$ (LSE) of the logits $z$.

$$\mathrm{sharpness}(z)_{\mathrm{DTKD}} = \log\left(\sum_i \exp(z_i)\right) \tag{14}$$

However, using LSE as the sharpness measure excludes case (3) because it is affected by magnitude, not correctness: The LSE function captures the overall magnitude of the logits, but does not directly account for whether the prediction is correct. **We claim that a better measure of sharpness would be the softmax probability at the true label**.

$$\text{sharpness}(z)^m_{True} = P^m_{z_T} = \frac{\exp(z_T)^m}{\sum_i \exp(z_i)^m} \qquad for \quad i \in C \tag{15}$$

T is the true label $y_k$. We can further normalize these values to obtain a relative ranking of a model m within our classroom $\mathcal{C}$.

$$\text{rank}(m) \propto \left( \frac{P^m_{z_T}}{\sum_{i \in \mathcal{C}} P^i_{z_T}} \right) \tag{16}$$

The above conditions can be reformulated as follows:

- $\alpha^m \propto \text{rank}(m)$

- $\alpha^m = \begin{cases} 0 & \text{if } \text{rank}(m) < \text{rank}(s) \\ \alpha^m & \text{otherwise} \end{cases}$

- $\tau^m \propto \frac{|P^m_{z_T} - P^s_{z_T}|}{P^m_{z_T}} \propto \frac{|\text{rank}(m) - \text{rank}(s)|}{\text{rank}(m)}$, given $\text{rank}(m) \neq 0$

where $\alpha^m$ and $\tau^m$ are the weight and temperature of KDLoss of student s with model m. This will now ensure that the student mostly learns from the correct classifiers. We say "mostly" because sometimes the wrong classifier can be sharper than the student at the true label (eg. green mentor in case-3). However, with our selective distillation method, we assume that the student will soon learn to filter out such classifiers.

**Adaptive $\tau$:** From equation 8, the temperature of the teacher is:

$$\tau_t = 1.0 + \frac{|R^t - R^s|}{R^t} \cdot \tau$$
$$= 1.0 + \frac{(P^t_{z_T} - P^s_{z_T})}{P^t_{z_T}} \cdot \tau \qquad \text{given} \quad R^t \geq R^s$$

When the difference between the student-teacher performance is very high, $P^t \gg P^s$, we get the max value $(\tau_t)_{max} = 1.0 + \tau$, the highest amount of simplification and when they are almost equal, $P^t \approx P^s$, we have the lowest $(\tau_t)_{min} = 1.0$ with no softening.

### F.2.1 DYNAMIC CAPACITY GAP

To better understand probabilistic distributions (Fig. 12) of our classroom, we plot the softmax of the logits produced by the student model and mentors at various training steps.

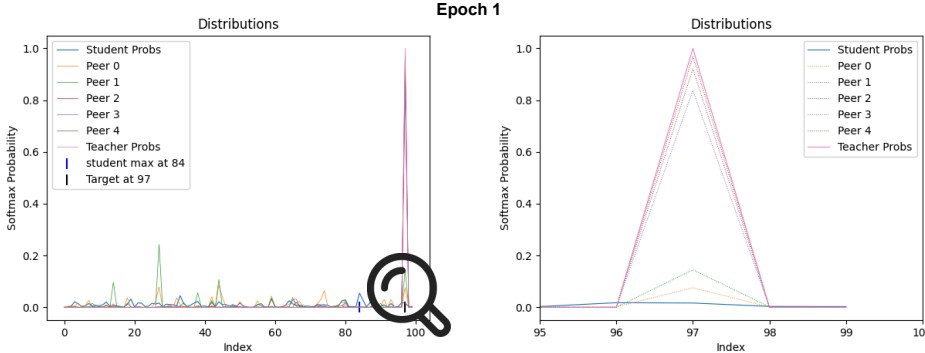

**Figure 13:** Probability Distributions at Epoch 1. Right subplot is zoomed in at the true class (97).

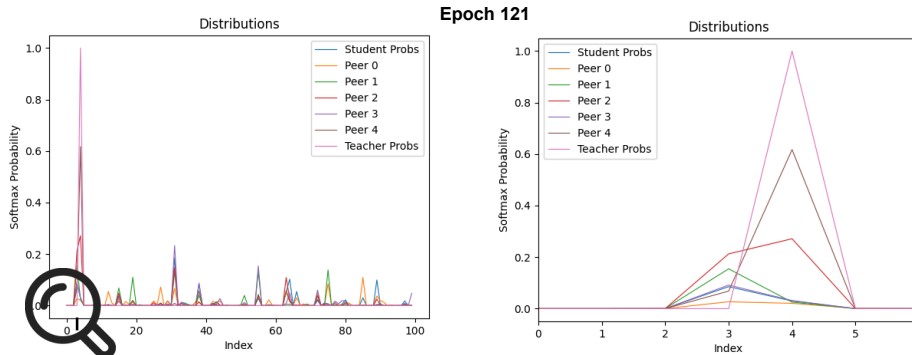

**Figure 14:** Probability Distributions at Epoch 121. Right subplot is zoomed in at the true class (4)

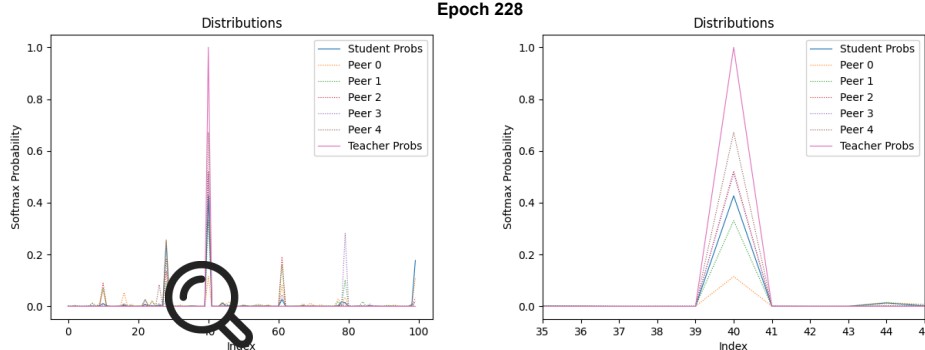

**Figure 15:** Probability Distributions at Epoch 228. Right subplot is zoomed in at the true class (40)

We observe a gradual decrease in the gap between the student and teacher's probabilities at the true label from epochs 1 through 228.

In Fig. 13, the "gap" between the strongest mentors and the student is higher compared to the weaker teachers, indicating the degree of softness applied to stronger teachers should be higher.

In Fig. 14, notice the "confusion" caused by classifiers who wrongly predict class 3. If not filtered out, these classifiers can pull the student's probability distribution in an undesirable direction. This is an example of case (3). However, in our method, these mentors' loss will be weighted low, proportional to their true label probability.

By the end of epoch 228 in Fig. 15, the student has successfully learned to filter out those mentors and mimic the teacher.

### F.3 CLASSROOMKD FOR 2D HUMAN POSE ESTIMATION

The proposed methodology can be applied to distill knowledge to smaller models in 2D HPE with a few modifications.

#### F.3.1 TOP-DOWN SIMCC-BASED METHODS

RTMPose architecture, which we use for our experiments on the COCO Kepoints dataset, contains a SimCC (Li et al., 2021a) head that outputs separate logits of the shape (N, K, D) each in the x and y directions, where N is the batch size, K is the number of joints, and D is the coordinate dimensions. For our purposes, only K is relevant. This output can be seen as **two predictions** for each of the K joints. Hence, we apply the following three modifications to adapt our approach:

1. The sharpness of model m, $P^m$, is calculated using the PCK accuracy metric. These values are further normalized in the classroom to obtain their respective ranks.

2. Once the *active* mentors are chosen, the $\mathcal{L}_{\text{distill}}$ is processed as the combined distillation loss between the student and mentor along x and y directions.

3. The logits' shapes are converted to (N*K,-1) before applying the KL-divergence. The sum of distillation losses along the x and y directions is finally divided by the number of joints.

$$\mathcal{L}_{\text{simcc}}(\hat{\boldsymbol{y}}^m, \hat{\boldsymbol{y}}^s; \tau^m) = \frac{1}{K}(\mathcal{L}_{\text{distill}}(\hat{\boldsymbol{y}}_x^m, \hat{\boldsymbol{y}}_x^s; \tau^m) + \mathcal{L}_{\text{distill}}(\hat{\boldsymbol{y}}_y^m, \hat{\boldsymbol{y}}_y^s; \tau^m)) \quad (17)$$

### F.3.2  TOP-DOWN HEATMAP-BASED METHODS

The LiteHRNet model, which we use for our experiments on the MPII Human Pose dataset, outputs 2D heatmaps of size (N, K, H, W). This is equivalent to the two separate 1D heatmaps in SimCC heads. To apply ClassroomKD in this case, we make the below changes:

1. Similar to the SimCC head, the sharpness of model m, $P^m$, is calculated using the PCK metric for the ranking.

2. The KL-divergence between the student and *active* mentors is calculated between the heatmaps and is then divided by the number of joints.

