# OpenReview forum: "Classroom-Inspired Multi-Mentor Distillation with Adaptive Learning Strategies"
_ICLR.cc/2025/Conference — Submitted to ICLR 2025_

### Official Review · Reviewer_gtoW · 2024-11-03

**Soundness:** 4
**Presentation:** 4
**Contribution:** 3
**Rating:** 5
**Confidence:** 4

**Summary:**

This paper presents a knowledge distillation framework ClassroomKD to improve knowledge transfer from multiple mentors to a student model by dynamically selecting mentors based on their effectiveness for each data sample. The framework includes a Knowledge Filtering Module, which ranks and activates high-quality mentors, and a Mentoring Module, which adjusts each mentor's influence according to the performance gap with the student. Experiments on CIFAR-100, ImageNet, COCO Keypoints, and MPII Human Pose indicate that ClassroomKD performs competitively with existing methods, suggesting that adaptive mentor selection can enhance knowledge transfer and model performance.

**Strengths:**

1. This paper provides an in-depth analysis of existing methods and highlights their respective limitations.
2. The proposed framework is effective, and some experimental results look good.
3. The paper is well-written and easy to understand.

**Weaknesses:**

1. In the second section, recent literatures on knowledge distillation from the past two years is limited, and it is recommended to include additional references. For example: [1] Logit standardization in knowledge distillation  [2] Class attention transfer based knowledge distillation.
2. In Figure 2, based on the article's content, the indicator label for Active mentors should be "M’"
3. According to the results in Table 1, the improvement achieved by the proposed method compared with the single-teacher distillation methods appears modest. Theoretically, the use of multiple mentors should yield a more significant improvement than a single mentor. It is recommended to conduct further experimental analysis to explore this aspect.

**Questions:**

Please refer to the Strengths and Weaknesses.

---

> ### Author Response · Authors · 2024-11-18
>
> Thank you for your thorough review and constructive feedback on our submission. We greatly appreciate your positive evaluation of our framework's **soundness, presentation, and contributions**, as well as your recognition of its **effectiveness and clear writing**. Below, we address the weaknesses and questions raised in your review.
>
> - **Inclusion of Recent Literature on Knowledge Distillation**: We appreciate your suggestion to include more recent works, such as [1] and [2], and agree that this will provide a more comprehensive view of the field. In the revised version of the paper, we will incorporate these references and additional recent works in Section 2 and highlight how our method relates to these approaches.
>
> - **Correction of Notation in Figure 2**: Thank you for pointing out the mismatch in the notation for active mentors in Figure 2. We have updated the figure and the corresponding text to consistently use "M’" to denote active mentors.
>
> - **Modest Improvements Compared to Single-Teacher KD**: We agree with the reviewer that, theoretically, multi-teacher methods should yield better performance than single-teacher distillation. However, in practice, this expectation does not always hold, especially when more than two teachers are used. This is due to several practical challenges, including error accumulation, lack of dynamic adaptation, and the increasing capacity gap between the teachers and the student model. These effects have been documented in existing research on multiple-teacher distillation, such as TAKD and DGKD. To address this point in more detail:
>   - **Performance Gap with Naive Multi-Teacher Distillation**: We would like to refer the reviewer to Figure 4(c), where we compare ClassroomKD with naive multi-teacher distillation (AVER). This figure demonstrates an increasing performance gap between ClassroomKD and AVER as the number of mentors increases from 4 to 6. This trend highlights the key issue: when using a larger number of mentors, naive distillation approaches such as AVER struggle to handle the conflicting and redundant information from multiple mentors, leading to suboptimal performance.
>   - **Comparison with Single-Teacher Methods**: Additionally, we point out that even existing state-of-the-art multi-teacher distillation methods (e.g., AVER, TAKD, DGKD) do not always outperform the single-teacher scenario. For example, in Table 1, the R20 student with R56 teacher achieves 72.05% accuracy with single-teacher DTKD, which is higher than the performance of all multi-teacher distillation methods in Table 2, except for ClassroomKD (which has 72.65%). This suggests that the theoretical expectation of multi-teacher methods consistently outperforming single-teacher methods does not always translate into practice, reinforcing the importance of methods like ClassroomKD to overcome these limitations.
>   - **Effect of Capacity Gap**: The modest improvements in some scenarios are also influenced by the increasing capacity gap between the mentors and the student model, which limits the amount of knowledge that can be effectively transferred. Additionally, naive ensemble methods often fail to adapt dynamically to varying teacher qualities, resulting in suboptimal distillation compared to a carefully tuned single-teacher KD setup.
>
> We hope these clarifications address the reviewer’s concerns regarding the observed improvements and provide insights into the theoretical and practical nuances of multi-teacher distillation. Thank you again for raising this important point, which allowed us to strengthen the discussion in our revised manuscript.

---

### Official Review · Reviewer_SG8E · 2024-11-03

**Soundness:** 2
**Presentation:** 3
**Contribution:** 2
**Rating:** 3
**Confidence:** 5

**Summary:**

This work deals with a multi-teacher knowledge distillation method, where the key idea is to assess the fitness of each teacher at individual training sample level, and use that fitness to select the teachers for knowledge distillation in a weighted manner. The proposed method has been evaluated on a number of image classification tasks in comparison to previous methods.

**Strengths:**

Clear writing up and presentation.

Good coverage of the literature.

Reasonably good experiment design and setup and presentation.

**Weaknesses:**

**Novelty**:
1) This data sample conditioned teacher weighting idea has been proposed in [Ref-A] where even more advanced meta-learning based optimization algorithm was proposed for model training - bilevel optimization algorithm, in addition to other complexity and challenges, like data access to other domains and label scarcity. In general, this proposed work is a subset of Ref-A in techniques.
- [Ref-A] Wang Z, Ye M, Zhu X, Peng L, Tian L, Zhu Y. Metateacher: Coordinating multi-model domain adaptation for medical image classification. Advances in Neural Information Processing Systems. 2022 Dec 6

2) More justification should be added on why only those mentors/teachers with higher probability estimation on the true class label are selected and activated for distillation, whilst the remaining teachers are not. For example, in cases that the student model makes too confident precision for a specific training sample, those less confident teachers may provide a signal to soften this predictive confidence. This is because, in knowledge distillation, the key knowledge with the teacher is mostly about the class distribution, rather than seeking the maximum of true class probability. Under this consideration, the proposed knowledge filtering design is questionable.

3) Similarly, more discussion and explanation about how Eq (8) represents the performance gap should be made. The similar concern holds here as the above.

**Experiments**:
1) The performance gap in comparison to previous art knowledge distillation is some limited, mostly within 0.5%. This suggests the benefits of this method is not significant.

**Presentation issues**:
1) In general, the first appearance of previous work should come with a reference, e.g., no reference for DGKD (Line 50), and no reference for those mentioned methods in Fig 1's caption. The authors need to take a careful global check for this.

2) The two concepts, performance gap (Line 52) and capacity gap (Ling 40), seem mixed and they are not properly defined, discussed and compared in the Introduction.

**Overall**:
Given the limited novelty in terms of techniques, lacking of solid design rationales,  and not significant experimental advantage, I do not find enough significance of accepting this work at the current shape. More research works are needed to further enhance it for future submission.

**Questions:**

Please check the weaknesses above.

---

> ### Author Response · Authors · 2024-11-18
>
> We sincerely thank the reviewer for valuable feedback. We hope to address each of the issues below:
>
> ### Novelty
>
> **W1: Metateacher [Ref-A]**
>
> - Metateacher’s [Ref-A] core ranking approach is similar to that of [A], as suggested by reviewer yecG. They use two learnable parameters to create a single weighted soft label from all teachers and then apply the KD loss. Another difference is that they use intermediate features of the student network.
> - Moreover, Metateacher is proposed as a domain-specific method for medical image classification. In contrast, our method is a general-purpose multi-teacher KD method, which we benchmark on standard datasets following established KD protocols.
> - Our proposed method distills separately from each mentor whose losses are weighted with their respective ranks. Our main novelty is the unification of weighting and mentorship (the two hyperparameters in logit-based KD) based on classroom-inspired dynamics. Although multi-teacher KD research revolves mainly around varying weights, mentorship has been researched mainly in single-teacher frameworks (DTKD, CTKD).
>
> We thank the reviewer for bringing this to our attention and will include a discussion in the revised version.
>
> **W2: Knowledge Filtering Design**
>
> We would like to clarify that, in our method, we choose mentors that are **both correct** and **more confident** than the student in predicting the true label using the masking in Eq. 3 and the ranking in Eq. 5. This implies:
>
> > If the student (or any other model) makes a wrong prediction with high confidence, this would result in low confidence on the true label and therefore a low rank.
>
> If the student and the mentor are **both correct**, but the **student is more confident**, the mentor is not selected for distillation. This avoids moving the student's probability distribution in the wrong direction, as explained in Appendix F.2 of our updated paper PDF.
>
> Our method does **not change the goal of KD**. The objective is still to bring the student's probability distribution closer to that of the teachers. However, we only minimize the KL divergence with teachers whose distributions are more representative for the true label, ensuring the student's performance is not negatively affected.
>
> We argue that **naive averaging** of all teachers, as done in AVER, can harm the student’s performance. For instance, in Tab. 2, SN-V1 with AVER KD achieves 73.00% accuracy, compared to 74.83% with single-teacher KD in Tab. 1. This effect is especially pronounced in heterogeneous architectures, where the teachers’ probability distributions vary significantly.
>
> Selecting only the most relevant teachers ensures the student maximizes its confidence on the true label without being distracted by less reliable signals. Our experimental results (Tab. 2 and Appendix Tab. 9) show that this approach consistently outperforms baselines. This demonstrates the practical value of our ranking scheme, particularly in multi-teacher distillation.
>
> **W3: Mentoring Module** The mentorship formula deals with the dynamic capacity(or performance) gap between the student and mentor(peer and teacher). This dynamic nature is illustrated in the appendix plots. We assume that the peers have a shallow understanding of the distribution while the teacher has a more detailed understanding. Distilling directly from the teacher during the start of the training is not very useful for the student. Hence, the peers aid it better without softening. As the training continues and the student-mentor gap reduces, the student can grasp the detailed explanation by the teacher, and hence, it is less softened. This is captured by Eq. 8 by providing maximum support to the student at the start and slowly scaffolding (similar to a popular neuroscience theory Zone of proximal development (Lev Vygotsku, 1978) which suggests that students learn best when interacting with peers and teachers who can scaffold their understanding.).
>
> ### Experiments
> **W1: Marginal improvements:** As presented in Tab. 9 in the updated appendix, our method shows significant gains over KD and AVER consistently for all student-mentor(s) configurations, unlike other methods. These values range between (+0.22 to +2.8) and (+1.25 to +7.01) respectively. We would also request the reviewer to see our response to Reviewer yecG regarding the same issue.
>
> ### Presentation Issues
> Thank you for bringing these issues to our attention. We acknowledge the confusion caused by using the two terms interchangeably. We will clarify this in the revised version. However, in the specific line (L52), the non-static "performance gap" applies both to the capacity and the difference in model accuracies (performance). We will make these changes in the revised version.
>
> ---
> We hope our responses have addressed their concerns. We look forward to engaging in further discussion. We would like to request the reviewer to also see Appendix F.2 and consider our work for acceptance.

---

### Official Review · Reviewer_6R5A · 2024-11-04

**Soundness:** 4
**Presentation:** 4
**Contribution:** 3
**Rating:** 6
**Confidence:** 3

**Summary:**

The paper presents ClassroomKD, a novel multi-mentor knowledge distillation framework inspired by classroom dynamics. It addresses challenges in multi-mentor distillation and consists of a Knowledge Filtering Module and a Mentoring Module. Experiments on multiple datasets show its superiority over existing methods.

**Strengths:**

1. The framework's dynamic mentor selection and adaptive teaching strategies are highly innovative. It simulates a classroom environment, a novel approach compared to traditional methods, and effectively solves problems in multi-mentor distillation.
2. Theories behind the KF and Mentoring Modules are reasonable. The loss function construction is also sound. Experiments on diverse datasets with detailed settings and in-depth result analysis provide strong evidence for the method's effectiveness. It contributes to knowledge distillation research, inspiring future work. Its good performance in computer vision tasks offers practical solutions.
4. The paper has a clear structure and logical flow. The writing is clear, and figures aid understanding. The appendix enriches the paper.

**Weaknesses:**

1. In-depth Analysis of Limitations: The limitations section could be enhanced. For example, more details on challenges in applying to other domains and the impact of framework complexity on performance and tuning difficulties are needed.
2. Exploration of More Practical Application Scenarios: While successful in computer vision, its potential in other areas like NLP and recommendation systems should be explored to show broader applicability.
3. Sensitivity Analysis of Hyperparameter Settings: A sensitivity analysis of hyperparameters would help researchers better understand and apply the method.
Overall, the paper has many strengths but could be improved in the mentioned areas.

**Questions:**

see the weaknesses

---

> ### Author Response · Authors · 2024-11-18
>
> Thank you for your thorough review and valuable feedback on our work. We greatly appreciate your recognition of the **novelty and effectiveness of our proposed ClassroomKD framework** and your positive comments on the **clarity of the paper**, the **robustness of our experiments**, and the **soundness of the methodology**.
>
> We have carefully considered the weaknesses and questions you raised, and we address them in detail below:
>
> - **W1: In-depth Analysis of Limitations**
> We agree that elaborating on the limitations could enhance the paper. While our current limitations section  (L535-539) briefly discusses the applicability to other domains, we will expand it in a revised version to explicitly address:
>   - Challenges in applying ClassroomKD to domains outside computer vision, particularly those with fundamentally different data structures or objectives.
>   - The framework's complexity, particularly regarding dynamic mentor selection and adaptive mentoring. While these components contribute to performance gains, we acknowledge that they could require careful tuning, which might pose challenges for practitioners. In the revised version, we will include a discussion on practical strategies to simplify implementation or mitigate complexity.
>
> - **W2: Exploration of Broader Applicability**
> As is standard in most knowledge distillation (KD) literature, we focused on computer vision tasks to maintain comparability with prior work. Our experiments span diverse datasets and tasks (e.g., CIFAR-100 classification, ImageNet, and 2D human pose estimation), demonstrating the robustness of ClassroomKD across varying complexities and data distributions. We appreciate your suggestion to explore broader domains, and while that is beyond the scope of the current work, it presents an exciting avenue for future research.
>
> - **W3: Sensitivity Analysis of Hyperparameters**:
> Thank you for highlighting this point. As shown in Figure 3, we performed a grid search for the key hyperparameter τ, which governs the temperature in the mentoring module. The results clearly demonstrate that τ=12 yields optimal performance for classification tasks. For pose estimation tasks, we selected task-specific values (τ=4) based on validation performance. We will further clarify this process in the revised manuscript. Additionally, we will add a discussion in the appendix on other hyperparameters (e.g., β, λ) and their influence on the overall performance to assist researchers in effectively applying the framework.
>
> We hope our responses address your concerns and demonstrate our commitment to strengthening the paper. Given the novelty of ClassroomKD, its strong empirical validation on diverse computer vision tasks, and its practical contributions to the KD field, we kindly request you to consider revising your rating to reflect the broader significance and impact of this work.
>
> Once again, thank you for your constructive feedback and insightful suggestions, which have been invaluable in helping us improve the manuscript.

---

### Official Review · Reviewer_yecG · 2024-11-05

**Soundness:** 2
**Presentation:** 2
**Contribution:** 2
**Rating:** 3
**Confidence:** 4

**Summary:**

after reading the comments from other reviewers and corresponding responses, I would like to keep my initial score due to limited novelty and insufficient experiments.

————
This paper follows a setting in knowledge distillation where there are multiple teacher models involved. Instead of using traditional approach that all teacher models influence the student model, it proposes to first rank the performance and then filter out teacher models that underperform the student one. Afterwards, use a KL divergence driven weight assignment according to the performance gap.

Experiments are conducted on CIFAR100, ImageNet and COCO. However, one thing to notice that on ImageNet and COCO pose estimation tasks, the authors only compare their method with the baseline student method without KD (NOKD).

**Strengths:**

+ The motivation that let more models teach the student model is not novel, but their approach to dynamically rank and select the better teachers per sample seems interesting and novel to me.

+ The paper is clearly written and easy to follow. The figure of motivation clearly shows the differences between their approach with earlier works.

+ Experiments on CIFAR100 are very comprehensive.

+ Ablation studies are complete and cover many aspects of the proposed ClassroomKD.

**Weaknesses:**

I have the following concerns and hope to see the response from the authors.

1. How does the ClassroomKD compare to other multiple teachers approaches? For example, [A] proposes a dynamic framework that also learns from multiple teachers and multi-level knowledge. The proposed ranking the best seems very similar to the proposal in [A].

2. While experiments on CIFAR-100 are very comprehensive, the evaluation on ImageNet and COCO pose estimation are not sufficient. Could you clarify the meaning of the AVER performance in the two tables? Additionally, please explain the underlying method used for these experiments.

3. On CIFAR100, the performance gap between the proposed method and others is somewhat marginal. It may be not a good way to evaluate your method on CIFAR100 since it is considered overfitted in current research. Results on larger, more challenging datasets would provide more valuable insights into the effectiveness of your approach.

4. It would be beneficial to include comparisons with at least one state-of-the-art multi-teacher method on larger datasets such as ImageNet or COCO.


[A] Liu et al. Adaptive multi-teacher multi-level knowledge distillation. Neuralcomputing

**Questions:**

Please see the above weaknesses.

---

> ### Author Response · Authors · 2024-11-18
>
> Thank you for your detailed review and constructive feedback on our work. We are grateful for your acknowledgment of the novelty in our dynamic mentor ranking and adaptive teaching strategies, as well as your positive remarks on the comprehensive experiments, clear writing, and ablation studies.
>
> ---
> **Comparison with [A]:**  While we recognize some high-level similarities, our method introduces several key distinctions from [A]:
>
> - **Ranking Method:** Our approach dynamically ranks mentors per sample based on relative performance, ensuring students learn from the most relevant mentors. [A], on the other hand, combines all teacher outputs into a single soft target, which lacks our fine-grained, per-sample mentor selection. This distinction is evident in the results on CIFAR-100 with an R20 student (Stu1 in [A]):
>   | | Top-1 | $\Delta$ over NOKD |
>   |---|---|:---:|
>   | NOKD | 69.06 | - |
>   | [A] | 70.39 | 1.34 |
>   | Ours | 72.65 | 3.63 |
>
> - **Knowledge Type:** Unlike [A], which uses both logit-based and hint-based distillation, we focus solely on logit-based distillation, achieving superior performance while maintaining implementation simplicity. This approach is extensible to hint-based distillation as well.
>
> ---
> **Comparison to multi-teacher methods:** This is a central focus of our work and is extensively addressed:
> - **Algorithmic Comparison:** Multi-teacher methods are divided into online (e.g., DML, ONE, SHAKE) and offline (e.g., AVER, AEKD, TAKD, DGKD) approaches. Online methods involve mutual learning among models, while offline methods focus on unidirectional knowledge transfer. A detailed discussion of these categories is provided in Sec. 2.2, along with comparisons in Fig. 1 (a-d) and Sec. 1 (L047-052). Tab. 5 and L484-489 further discuss adaptive temperature approaches. Space limitations precluded deeper algorithmic details of existing methods in our paper.
>
> - Tab. 2 highlights ClassroomKD's performance against **nine multi-teacher methods**, demonstrating consistent superiority.
>
> ---
> **Definition of AVER:** This simple multi-teacher baseline assumes equal weighting for all teachers, modifying Eq. 10 to $L_{AVER}= L_{task} + \sum_{m \in M} KL (s \parallel m)$. This approach serves as the multi-teacher equivalent of baseline KD for single teachers, aligning with SOTA works like SHAKE and CA-MKD, who also use AVER as their baseline.
>
> ---
> **CIFAR-100 Results**
>
> - **Dataset Choice:** CIFAR-100 remains the standard dataset for benchmarking KD methods in SOTA research because of its manageable scale and widespread use. It is widely used in KD research, enabling fair and direct comparisons. Using this dataset allows us to benchmark against existing works without independently retraining each method.
>
> - **Larger Datasets:** Some recent works also provide preliminary results on ImageNet, but the student/teacher pairs used are often inconsistent, making comparisons difficult. While we recognize the value of results on larger datasets and provide a comparison of our ClassroomKD with baseline multi-teacher KD (i.e., AVER), we cannot compare with other baselines because (1) none of them provide results using the same student as us, and (2) training on ImageNet takes several days. However, our method consistently outperforms baseline multi-teacher distillation on ImageNet classification and COCO key points estimation (Tab 2 (b-c)). We aim to expand this in future work.
>
> - **Performance Margins:** Gains on CIFAR-100 may appear small, but they are consistent across **12 different architectures**, **four datasets**, and **two vision tasks**, reflecting ClassroomKD's robustness. Tab. 9 in the updated appendix shows gains over AVER range from +1.25 to +7.01. Notably, we observe superior generalization in larger classrooms and diverse mentor compositions (see ablation studies).
>
> ---
> For pose estimation on **MPII** and **COCO** datasets, we use the PCK metric to rank mentors instead of true label probability. The task-specific MSE loss is used in Eq. 10 (also see L265). The overall methodology remains unchanged.
>
> [1] SimCC: A Simple Coordinate Classification Perspective for Human Pose Estimation
>
> ---
> We hope our responses address your concerns and clarify the novelty, effectiveness, and robustness of ClassroomKD. Given the strengths of our work, we kindly request you to consider raising your score to recommend acceptance. Your thoughtful review and constructive suggestions have been invaluable, and we are committed to further refining the paper based on this feedback.

---

### Meta-Review · Area_Chair_C4qh · 2024-12-19

**Metareview:**

This paper introduces a multi-mentor knowledge distillation framework known as ClassroomKD. It includes a Knowledge Filtering Module that ranks and activates high-quality mentors and a Mentoring Module that adjusts each mentor's influence based on the performance gap with the student. Experiments conducted on CIFAR-100, ImageNet, COCO Keypoints, and MPII Human Pose demonstrate that ClassroomKD performs competitively compared to existing methods.

The paper received mixed reviews, with scores of 6, 5, 3, and 3, leading to an average score of 4.25. Despite a rebuttal, concerns regarding its limited technical novelty and insufficient experimental validation remain unaddressed. Therefore, the Area Chair recommends rejection at this time.

**Additional Comments On Reviewer Discussion:**

The issues related to novelty and inadequate experimental validation are not adequately addressed.

---

### Decision · Program_Chairs · 2025-01-22

Reject